# ALKBH7 mediates necrosis via rewiring of glyoxal metabolism

**Chaitanya A Kulkarni[1], Sergiy M Nadtochiy[1], Leslie Kennedy[2], Jimmy Zhang[1], Sophea Chhim[3], Hanan Alwaseem[4], Elizabeth Murphy[2], Dragony Fu[3], Paul S Brookes[1]\***

[1]Department of Anesthesiology & Perioperative Medicine, University of Rochester Medical Center, Rochester, NY, United States; [2]NHLBI Intramural Research Program, National Institutes of Health, Bethesda, United States; [3]Department of Biology, University of Rochester, Rochester, NY, United States; [4]Department of Chemistry, University of Rochester, Rochester, NY, United States

**Abstract** Alkb homolog 7 (ALKBH7) is a mitochondrial α-ketoglutarate dioxygenase required for DNA alkylation-induced necrosis, but its function and substrates remain unclear. Herein, we show ALKBH7 regulates dialdehyde metabolism, which impacts the cardiac response to ischemia-reperfusion (IR) injury. Using a multi-omics approach, we find no evidence ALKBH7 functions as a prolyl-hydroxylase, but we do find $Alkbh7^{-/-}$ mice have elevated glyoxalase I (GLO-1), a dialdehyde detoxifying enzyme. Metabolic pathways related to the glycolytic by-product methylglyoxal (MGO) are rewired in $Alkbh7^{-/-}$ mice, along with elevated levels of MGO protein adducts. Despite greater glycative stress, hearts from $Alkbh7^{-/-}$ mice are protected against IR injury, in a manner blocked by GLO-1 inhibition. Integrating these observations, we propose ALKBH7 regulates glyoxal metabolism, and that protection against necrosis and cardiac IR injury bought on by ALKBH7 deficiency originates from the signaling response to elevated MGO stress.

**\*For correspondence:**
paul_brookes@urmc.rochester.edu

**Competing interests:** The authors declare that no competing interests exist.

## Introduction

The α-ketoglutarate (α-KG) dioxygenases are a diverse enzyme superfamily, whose primary biochemical function is the addition of hydroxyl (–OH) to protein or nucleic acid substrates (*Islam et al., 2018*). The family includes the TET 5-methylcytosine hydroxylases, the EGLN prolyl-hydroxylases that regulate hypoxia-inducible factor (HIF), and the JmjC domain-containing histone demethylases. All α-KG dioxygenases use α-KG and $O_2$ as biochemical substrates and generate succinate as product. The AlkB homologs (ALKBHs) are a distinct family of nine α-KG dioxygenases that are homologs of *E. coli* AlkB (*Fedeles et al., 2015*). The bacterial AlkB enzyme catalyzes demethylation of DNA damaged by alkylating agents, via hydroxylation of the methylated DNA followed by spontaneous decomposition to release formaldehyde and recover the DNA base (*Falnes et al., 2002*; *Trewick et al., 2002*). Many eukaryotic ALKBHs have been shown to act on DNA or RNA substrates, including mammalian ALKBHs 1–3, 5, eight and FTO (*Aas et al., 2003*; *Duncan et al., 2002*; *Wu et al., 2016*; *Xie et al., 2018*). The subject of this investigation is ALKBH7, a poorly characterized mitochondrial α-KG dioxygenase which has no known substrates.

In contrast to a role in DNA repair or RNA modification, structural studies have revealed ALKBH7 lacks a critical nucleotide recognition lid required for binding DNA or RNA (*Wang et al., 2014*). Moreover, a study of mitochondria from several tissues of $Alkbh7^{-/-}$ mice showed no differences in mtDNA modifications (6-methyladenine, 5-methylcytosine, etc.) vs. wild-type (WT) at young ages, although $Alkbh7^{-/-}$ mtDNA did accumulate more modifications in old-age (*Pawar et al., 2018*). Together these observations suggest ALKBH7 may not play significant roles in either the repair of known AlkB substrates, or in oxidizing as-yet unknown nucleic acid substrates. Similarly, efforts to

identify potential protein substrates of ALKBH7 have not yielded insight to its function, with both a yeast-2-hybrid screen and several large mitochondrial protein:protein interaction databases not reporting any ALKBH7-binding proteins (*Bjørnstad et al., 2012*; *Schweppe et al., 2017*; *Liu et al., 2018*; *Floyd et al., 2016*).

While the nucleic acid or protein substrate(s) of ALKBH7 remain(s) unclear, previous studies using RNAi or genetic ablation in human cells have found that ALKBH7 is required for programmed necrosis induced by DNA alkylating agents (*Fu et al., 2013*). Moreover, *Alkbh7*$^{-/-}$ mice exhibit protection against alkylation-induced cell death in certain tissues (*Jordan et al., 2017*). Notably, this phenotype is only observed in males, even though *Alkbh7* is not a sex-linked gene. Furthermore, the *Alkbh7*$^{-/-}$ mouse exhibits obesity due to defective fatty acid β-oxidation (*Solberg et al., 2013*) and an *Alkbh7* mis-sense mutation (R191Q) has been linked to prostate cancer (*Walker et al., 2017*). However, it is unknown how these phenotypes are linked to the biochemical function of ALKBH7 as an α-KG dioxygenase.

The heart is a mitochondria-rich tissue, and cardiomyocyte necrosis plays a key role in cardiac pathology such as that occurring in ischemia-reperfusion (IR) injury (*Frangogiannis, 2015*). As such, the requirement for ALKBH7 in other models of necrosis (*Fu et al., 2013*) makes the protein a potential target for the modulation of cell death in response to IR injury. Herein, focusing on heart tissue we employed a multi-omics approach to elucidate ALKBH7 biology, finding that hearts from *Alkbh7*$^{-/-}$ mice are protected against IR injury. We also find that a core component of this protected phenotype is the rewiring of glucose and glyoxal metabolism in response to elevated glycative stress. These findings imply potential therapeutic utility for ALKBH7 inhibitors to prevent necrosis in IR and other conditions.

## Results

The complete original data used to generate all figures in the main document and supplement are contained in a spreadsheet available at DOI:10.6084/m9.figshare.12200273.

### Proteomic analysis to identify ALKBH7 substrates suggests it is not a prolyl-hydroxylase

Several members of the α-KG dioxygenase superfamily possess prolyl-hydroxylase activity, and ALKBH7 is known to auto-hydroxylate on Leucine 110, suggesting it has hydroxylase activity (*Wang et al., 2014*). To investigate the hypothesis that ALKBH7 may be a prolyl-hydroxylase, a tandem-mass-tag (TMT) proteomic approach was employed to identify potential targets, assuming such targets would contain less hydroxyproline (P-OH) in *Alkbh7*$^{-/-}$ vs. WT samples. Using heart tissue from *Alkbh7*$^{-/-}$ and WT mice (*Figure 1—figure supplement 1*), a total of 451 P-OH containing peptides were identified, and their abundances normalized to those of their 238 parent proteins. Differential analysis, applying thresholds of ±1.5 fold change and p<0.05 for significance, revealed only a handful of peptides with altered P-OH (volcano plot *Figure 1A*, top five up/down hits in *Figure 1B*). Only one peptide showed significantly less P-OH: the β-oxidation enzyme hydroxyacyl-CoA dehydrogenase (*Hadh* gene).

It is reported that *Alkbh7*$^{-/-}$ mice are obese and harbor a baseline defect in β-oxidation of long-chain fatty acids such as oleate, which can be overcome when stimulated by fasting (*Solberg et al., 2013*). To determine β-oxidation levels in the heart, a mostly fat-burning organ, Seahorse XF analysis of isolated cardiomyocytes from *Alkbh7*$^{-/-}$ and WT mice was undertaken, revealing a small but significant decrease in oleate oxidation at baseline, with this effect disappearing upon stimulation of maximal respiration (*Figure 1—figure supplement 2*). Although HADH is primarily involved in the β-oxidation of short chain fatty acids, we hypothesized based on P-OH proteomic data and the obese phenotype that prolyl-hydroxylation of HADH may regulate its activity. However, western blotting showed no alteration in HADH protein levels between WT and *Alkbh7*$^{-/-}$ (*Figure 1C*), and activity assays of both short-chain HADH (*Hadh* gene) and long-chain HADH (*Hadha* gene) revealed no differences between genotypes (*Figure 1D*). As such, we consider it unlikely that prolyl-hydroxylation by ALKBH7 is an underlying cause of defective β-oxidation in the *Alkbh7*$^{-/-}$ mouse.

To identify potential ALKBH7-binding partners, a FLAG-tag pull-down interactome experiment was performed, under either baseline or DNA alkylation stress conditions (see *Figure 2—figure supplement 1*). As the table in *Supplementary file 1* shows several mitochondrial heat-shock proteins

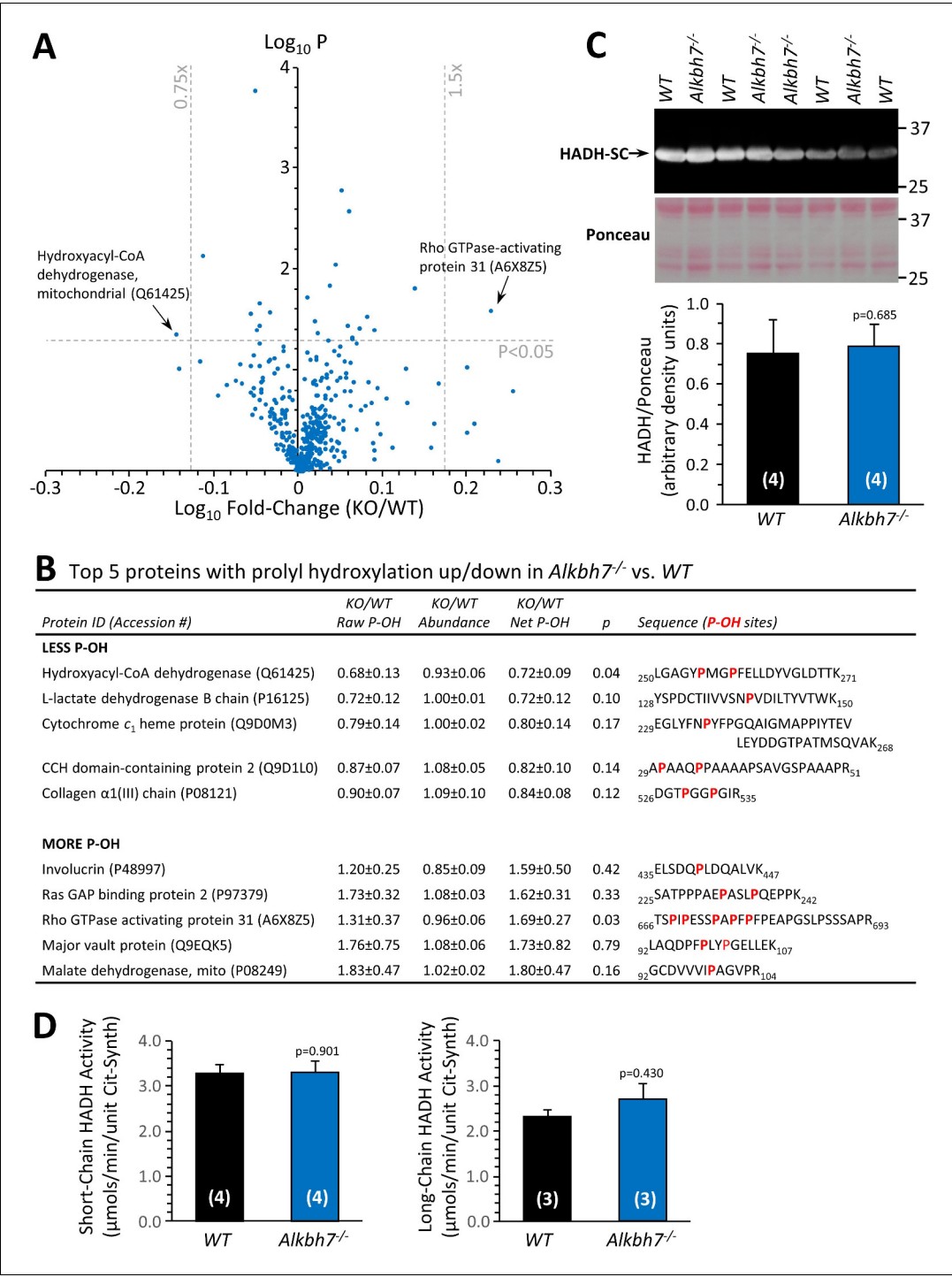

**Figure 1.** Proteomic analysis of prolyl-hydroxylation in WT vs. *Alkbh7*[-/-]. Hearts from young male WT or *Alkbh7*[-/-] mice were analyzed by tandem mass tag LC-MS/MS as per the methods. Abundance of each P-OH peptide was normalized to the abundance of its parent protein. (**A**) Volcano plot showing relative levels of 451 P-OH containing peptides. X-axis shows $Log_{10}$ of fold change (*Alkbh7*[-/-] / WT) and Y-axis shows $Log_{10}$ of significance (paired t-test, N = 5). Proteins crossing thresholds (gray lines) in upper left or right quadrants are labeled. (**B**) Table showing the top 5 P-OH containing peptides exhibiting increased or decreased relative abundance in *Alkbh7*[-/-] vs. WT. Table shows raw abundance of each P-OH peptide, abundance of the parent protein, and normalized abundance of the P-OH peptide. Annotated sequences highlight the hydroxylated proline residues in red. (**C**) Western blot showing abundance of HADH-SC (*Hadh*) in WT or *Alkbh7*[-/-] heart mitochondria with quantitation below, normalized to protein loading determined by Ponceau S stained membrane. (**D**) Spectrophotometric activity assays of short-

*Figure 1 continued on next page*

*Figure 1 continued*

chain and long-chain HADH in WT or *Alkbh7⁻/⁻* heart mitochondria. Bar graphs in panels C/D show means ± SE, N = 3–5, with p-values (paired t-test) shown above error bars. In bar graphs, N for each group is shown in parentheses.

The online version of this article includes the following figure supplement(s) for figure 1:

**Figure supplement 1.** Western blot showing absence of ALKBH7 protein in *Alkbh7⁻/⁻* mouse heart.

**Figure supplement 2.** Seahorse XF measurement of long chain fatty acid oxidation in WT vs. *Alkbh7⁻/⁻* cardiomyocytes.

---

were identified as ALKBH7 interactors, despite no such proteins being differentially hydroxylated (*Figure 1A*). This finding is in agreement with a recent antibody-based immunoprecipitation study which suggested a role for ALKBH7 in proteostasis (*Meng et al., 2019*), although the functional significance of this for necrosis is unclear (see *Figure 5—figure supplement 1* and related text). An additional protein hit was the NDUFS7 subunit of respiratory complex I, which the BioPlex interactome database also reports as an ALKBH7 interacting protein (*Schweppe et al., 2018*). However, enzyme assays in heart and liver mitochondria from WT and *Alkbh7⁻/⁻* mice revealed no differences in the activities of complex I and several other key mitochondrial enzymes (*Figure 2—figure supplement 2*), suggesting no role for ALKBH7 in regulating complex I function. Overall, consistent with a general paucity of ALKBH7-binding proteins (*Bjørnstad et al., 2012*; *Schweppe et al., 2017*; *Liu et al., 2018*; *Floyd et al., 2016*), we consider it unlikely that the necrosis function of ALKBH7 is due to prolyl-hydroxylation or the binding and modulation of mitochondrial heat shock proteins or respiratory complexes.

## Proteomic abundance analysis in Alkbh7⁻/⁻ indicates re-wiring of glyoxal metabolism

In parallel with analysis of P-OH, the TMT proteomic experiment also yielded relative abundance values for 3737 proteins in *Alkbh7⁻/⁻* and WT hearts, with a volcano plot (*Figure 2A*) revealing several differences which may underlie the metabolic phenotype of the knockout animals. Although ALKBH7 itself appears in the proteomic data set, this is not an indication of improper deletion. The original knockout targeted exons 2–5 containing the active site, whereas the peptides found here were in exon 1. While we cannot exclude the possibility of dominant negative effects due to an inactive truncation product, limited experiments with heterozygous animals (not shown) did not reproduce any phenotypes observed in homozygous knockouts. The lipid droplet protein perilipin-5, which signals via Sirt1/PPAR-α to drive mitochondrial biogenesis and fat oxidation (*Najt et al., 2020*), was 25% lower in *Alkbh7⁻/⁻* vs. WT. In addition, fructose-1,6-bisphosphatase 2 (FBP2/PFK2) was 31% lower in *Alkbh7⁻/⁻* vs. WT, a finding typically associated with acceleration of glycolysis (*Li et al., 2013*). It was also recently found that a mutation in ALKBH7 is selected for in the evolution of bats toward a nectivorous (sugar-rich) diet (*Gutiérrez-Guerrero et al., 2020*), suggesting that the enzyme is inherently linked to glucose metabolism.

In addition, a highly significant (p=0.00006) 1.6-fold elevation was seen in lactoylglutathione lyase (glyoxalase I, GLO-1) in *Alkbh7⁻/⁻* vs. WT. This observation was confirmed by enzymatic activity assay (*Figure 2B*) and by western blot (*Figure 2D/E*), with a similar activity difference also observed in *Alkbh7⁻/⁻* vs. WT liver tissue (*Figure 2—figure supplement 2*). GLO-1 is part of the dialdehyde detoxification pathway that handles toxic metabolites such as the glycolytic by-product methylglyoxal (MGO), recycling it to D-lactate, thus avoiding the generation of advanced glycation end products (*Figure 2F*; *Rabbani et al., 2016*). No change was seen in the activity of the companion enzyme GLO-2 in *Alkbh7⁻/⁻* (*Figure 2C*). The only other protein significantly upregulated in *Alkbh7⁻/⁻* was heme binding protein 1 (Hebp1), and notably a recent study found both GLO-1 and Hebp1 were upregulated in Alzheimer's disease (*Yagensky et al., 2019*), suggesting these proteins may share a common upstream regulator. Overall, despite extensive proteome coverage, a surprisingly small number of proteins (4) were up- or down-regulated in *Alkbh7⁻/⁻* heart.

Although ALKBH7 is generally thought to be mitochondrial, several of the differences observed between WT and *Alkbh7⁻/⁻* heart were cytosolic proteins, including GLO-1. In this regard, western blotting (*Figure 1—figure supplement 1*) showed immunoreactivity for ALKBH7 in the cytosolic

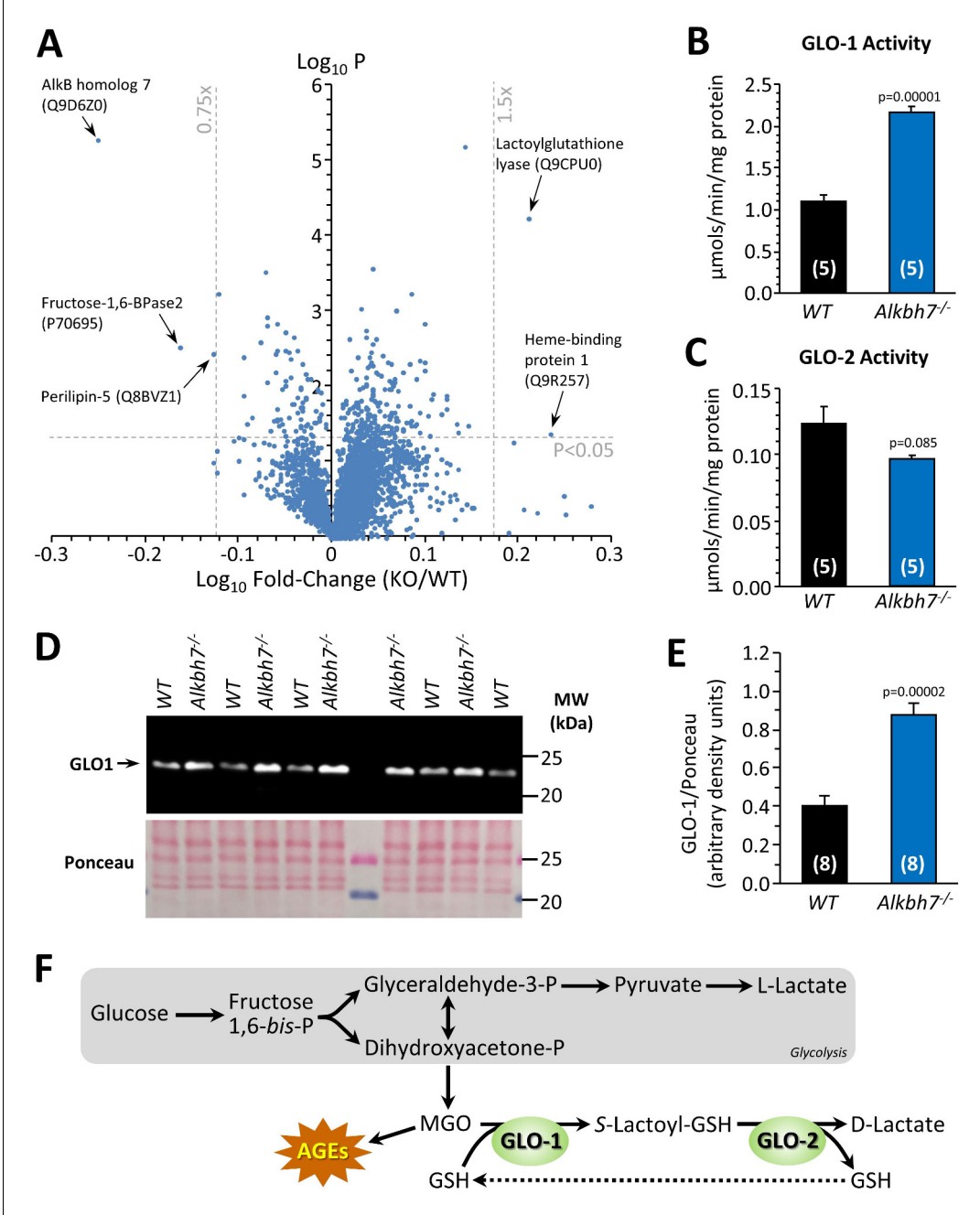

**Figure 2.** Proteomic analysis of protein abundance in WT vs.*Alkbh7⁻/⁻*. Hearts from young male WT or *Alkbh7⁻/⁻* mice were analyzed by tandem mass tag LC-MS/MS as per the methods. (**A**) Volcano plot showing relative levels of 3737 proteins. X-axis shows $Log_{10}$ of fold change (*Alkbh7⁻/⁻* / WT) and Y-axis shows $Log_{10}$ of significance (paired t-test, N = 5). Proteins crossing thresholds (gray lines) in upper left or right quadrants are labeled. (**B**) Activity of GLO-1 in WT or *Alkbh7⁻/⁻* heart cytosol. (**C**) Activity of GLO-2 in WT or *Alkbh7⁻/⁻* heart cytosol. (**D**) Western blot showing abundance of GLO-1 in WT or *Alkbh7⁻/⁻* heart cytosol, with Ponceau stained membrane below. (**E**) Quantitation of GLO-1 blot, normalized to protein loading. Bar graphs in panels B/C/E show means ± SE, N = 4–5, with p-values (paired t-test) shown above error bars. (**F**) Schematic showing the methylglyoxal detoxification system and its relationship to glycolysis. Abbreviations: AGEs: Advanced glycation end products, GSH: glutathione. MGO: methylglyoxal. In bar graphs, N for each group is shown in parentheses.
The online version of this article includes the following figure supplement(s) for figure 2:

**Figure supplement 1.** Representative gel from FLAG-tagged ALKBH7 pull-down experiment.
**Figure supplement 2.** Enzyme activities in WT and *Alkbh7⁻/⁻*tissues.

compartment (uncontaminated by the mitochondrial marker ANT-1) as well as in mitochondria, suggesting ALKBH7 may not be exclusively mitochondrial. The relative importance of sub-populations of ALKBH7 in driving necrosis or other phenotypes is currently unclear, and it is possible that differences in expression of cytosolic proteins may originate from the cytosolic population of the ALKBH7 enzyme.

## Metabolomics analysis in Alkbh7$^{-/-}$ confirms rewired glyoxal metabolism

To further probe metabolism in *Alkbh7$^{-/-}$* hearts, a steady-state metabolomics analysis was undertaken (*Figure 3A*), which revealed perturbations in several key metabolites related to MGO stress. The antioxidants carnosine and glutathione (GSH) were both significantly lower in *Alkbh7$^{-/-}$*, consistent with their being utilized in the detoxification of MGO (*Ghodsi and Kheirouri, 2018*). As noted above (*Figure 2F*), GLO-2 recycles GSH consumed by GLO-1, so an elevation in GLO-1 activity without a concomitant upregulation of GLO-2 would be predicted to result in GSH depletion. Furthermore, numerous metabolites in the lower half of glycolysis were elevated in *Alkbh7$^{-/-}$*, suggesting acceleration of this pathway (*Figure 3B*). To test this hypothesis directly, $^{13}$C-glucose tracing was employed to measure glycolytic flux in perfused mouse hearts (*Nadtochiy et al., 2015*), and the results in *Figure 3C* show that glycolytic flux was indeed faster in *Alkbh7$^{-/-}$*. While this might be anticipated to result in elevated MGO levels, unfortunately due to the labile nature of MGO attempts to measure free MGO levels in WT and *Alkbh7$^{-/-}$* samples were unsuccessful. However, further evidence for elevated MGO stress in *Alkbh7$^{-/-}$* was seen in the form of elevated MGO protein adduct levels in both cytosol and mitochondria (*Figure 3D–F*). Overall, these data suggest the *Alkbh7$^{-/-}$* heart experiences elevated glycative stress. Notably, neither α-ketoglutarate nor succinate (the respective substrate and product of α-KG dioxygenases) were significantly altered between WT and *Alkbh7$^{-/-}$*, indicating the enzyme does not contribute to the overall bulk turnover of these metabolites, relative to Krebs' cycle activity. This is in agreement with the finding that neither α-KGDH nor SDH activities were different between *Alkbh7$^{-/-}$* and WT (*Figure 2—figure supplement 2*).

## Loss of ALKBH7 protects the heart from ischemia-reperfusion (IR) injury

In addition to metabolic effects, a key phenotype resulting from *Alkbh7* ablation is protection against necrosis (*Fu et al., 2013*). In seeking links between glyoxal metabolism and necrosis, it is notable that both glycative stress and necrosis are implicated in the pathology of cardiac IR injury (*Almeida et al., 2013*; *Wang et al., 2010*; *Del Re et al., 2019*). In addition, a mitochondrially targeted MGO scavenging molecule was recently shown to protect the heart against IR injury (*Tate et al., 2019*). To test the hypothesis that loss of ALKBH7 may protect against IR, perfused hearts from WT and *Alkbh7$^{-/-}$* mice were subjected to IR injury (25 min global ischemia, 60 min reperfusion). As shown in *Figure 4A/B*, male *Alkbh7$^{-/-}$* hearts exhibited significantly improved post-ischemic functional recovery and significantly lower infarct size (an indicator of necrosis) compared to WT.

Consistent with sexual dimorphism in the necrosis effects of ALKBH7 (*Jordan et al., 2017*), no protection against IR injury was observed in hearts from female *Alkbh7$^{-/-}$* mice (*Figure 4—figure supplement 1A, B*). Interestingly, it was found that female *Alkbh7$^{-/-}$* mice still exhibited an elevation in GLO-1 at both the protein and activity level (*Figure 4—figure supplement 1C, D, E*). However, the elevated GLO-1 activity in female *Alkbh7$^{-/-}$* only approached the levels seen in WT males, owing to a lower baseline GLO-1 activity in females. Underlying this lower GLO-1 activity, we also found that that female *Alkbh7$^{-/-}$* mice did not exhibit an elevation in MGO adduct formation relative to WT (*Figure 4—figure supplement 1F, G, H*). This suggests that glycative stress is blunted in females, such that female *Alkbh7$^{-/-}$* mice may not reach a threshold of GLO-1 activity required for protection. Later results highlight the requirement of GLO-1 for protection of male *Alkbh7$^{-/-}$* mice.

Several paradigms of cardioprotection are known to decline with age (*Rahman et al., 2013*; *Boengler et al., 2009*; *Nguyen et al., 2008*; *Calabrese et al., 2015*), and consistent with this we also found that protection against IR injury was lost in aged male *Alkbh7$^{-/-}$* mice (*Figure 4—figure supplement 2*). Notably, GLO-1 activity has been shown to decline with age (*Rabbani et al., 2016*). Together, these results suggest that ALKBH7 may play a role in necrosis by regulating MGO metabolism, and its ablation triggers a signaling response that endows protection against IR in the heart.

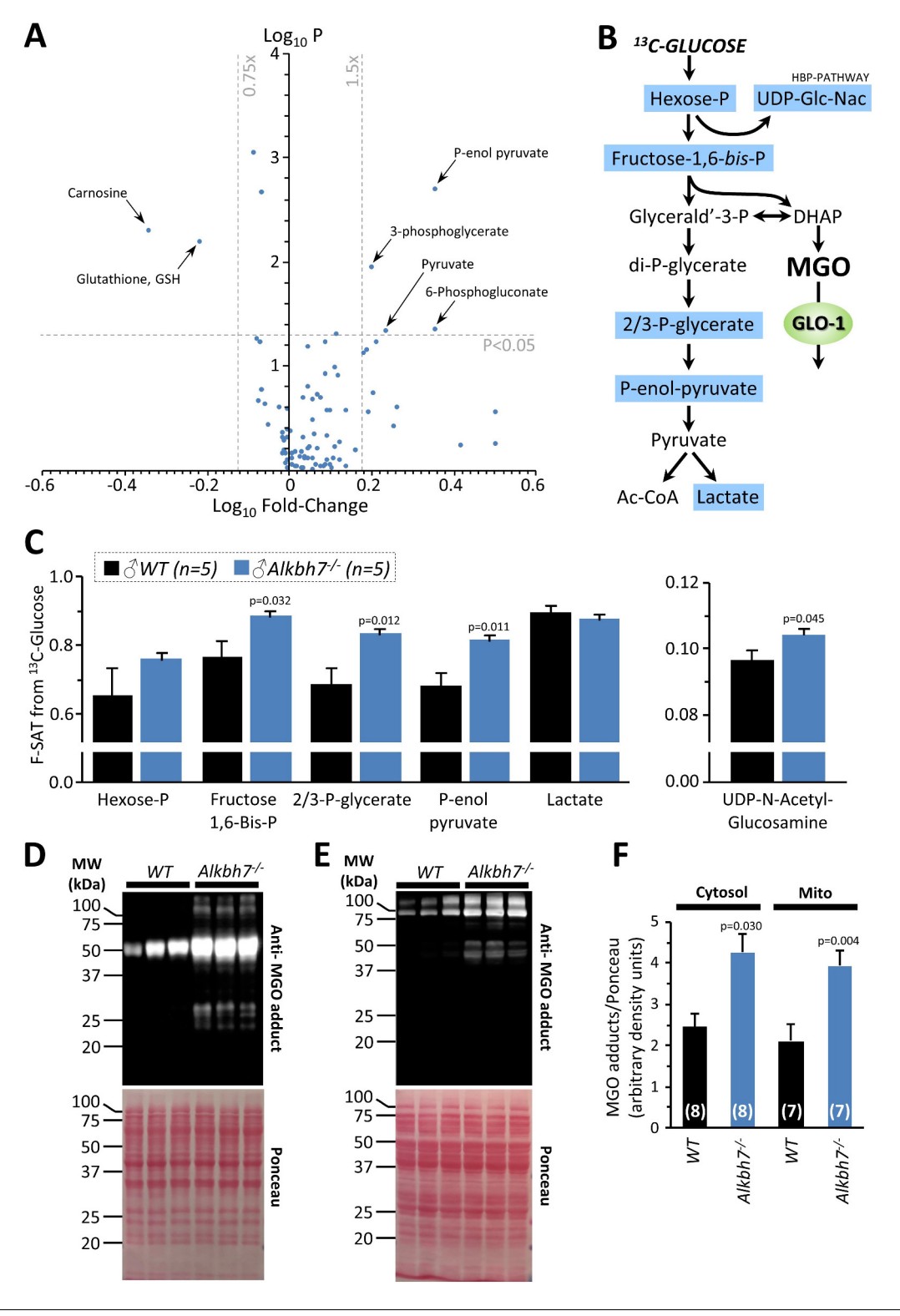

**Figure 3.** Metabolomics analysis in WT vs. *Alkbh7*[-/-]. Hearts from young male WT or *Alkbh7*[-/-] mice were analyzed by LC-MS/MS-based metabolomics as per the methods. (**A**) Volcano plot showing relative levels of 90 metabolites in the steady state. X-axis shows $Log_{10}$ of fold change (*Alkbh7*[-/-] / WT) and Y-axis shows $Log_{10}$ of significance (paired t-test, N = 8–17 depending on metabolite). Metabolites crossing thresholds (gray lines) in upper left or right quadrants are labeled. A pathway impact analysis is shown in ***Figure 3—figure supplement 1***. (**B**) Schematic

*Figure 3 continued on next page*

*Figure 3 continued*

showing glycolysis and its relationship to methylglyoxal (MGO). Metabolites quantified in $^{13}C$-flux measurements (panel **C**) are highlighted blue. (**C**) $^{13}C$-glucose flux measurements of glycolytic activity in *Alkbh7*$^{-/-}$ vs. WT hearts. Y-axis shows fractional saturation (F-SAT) of each metabolite within 5 min. from exogenously delivered [U-$^{13}C$] glucose. Note: UDP-Glc-Nac is shown on separate axes for clarity. (**D, E**) Western blot showing abundance of MGO-adducts in *Alkbh7*$^{-/-}$ and WT heart cytosol (**D**) or mitochondria (**E**). Ponceau stained membranes are shown below. (**F**) Quantitation of MGO adduct content from blots, normalized to protein loading. Bar graphs in panels C/ F show means ± SE, N = 4–5, with p-values (paired t-test) shown above error bars. In bar graphs, N for each group is shown in parentheses.

The online version of this article includes the following figure supplement(s) for figure 3:

**Figure supplement 1.** Representation of metabolic pathways based on detected metabolites.

## Acute pharmacologic ALKBH inhibition elicits cardioprotection

The activity of α-KG dioxygenases can be inhibited by D- or L- isomers of the non-canonical metabolite 2-hydroxyglutarate (2-HG) (*Evans et al., 2015*), with L-2-HG inhibiting ALKBHs more potently than D-2-HG (*Wang et al., 2015*; *Chen et al., 2017*). In addition, acute administration of the generic α-KG dioxygenase inhibitor dimethyloxalylglycine was shown to confer protection against hypoxic injury in a cardiomyocyte model of IR (*Sridharan et al., 2008*). Since genetic ablation of ALKBH7 was cardioprotective, we thus hypothesized its pharmacologic inhibition may serve a similar purpose. As *Figure 5—figure supplement 2* shows, administration of L-2-HG as its dimethyl ester (a common delivery strategy for dicarboxylates) was cardioprotective in WT hearts, eliciting enhanced functional recovery and lower infarct size (albeit the latter non-significant). While L-2-HG is known to have multiple targets, these data suggest the development of more specific ALKBH7 inhibitors may be a promising therapeutic strategy for IR injury.

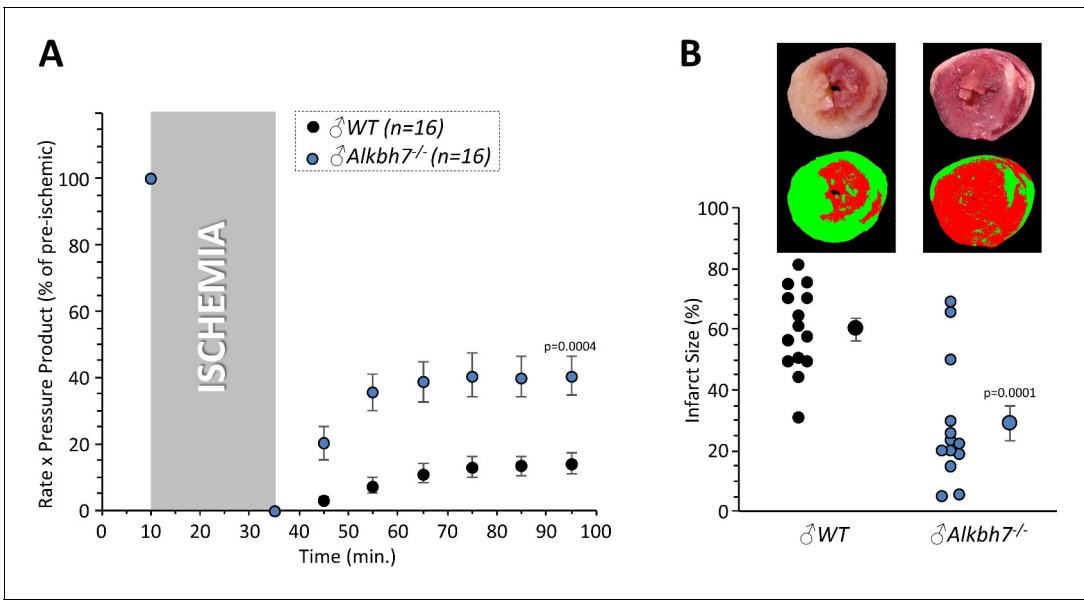

**Figure 4.** Response to ex-vivo cardiac ischemia-reperfusion (IR) injury in WT vs.*Alkbh7*$^{-/-}$. Hearts from young male WT and *Alkbh7*$^{-/-}$ mice were Langendorff perfused and subjected to 25 min ischemia plus 60 min reperfusion. (**A**) Cardiac function assessed by left ventricular balloon pressure transducer. Graph shows the product of heart rate multiplied by left ventricular developed pressure, as a percentage of the initial (pre-ischemic) value. (**B**) Post IR staining with TTC for quantitation of myocardial infarct size. Representative TTC-stained heart slices are shown, with pseudo-colored mask images used for quantitation by planimetry (red = live tissue, green = infarct). Data are quantified below, with individual data points to shown N, and means ± SE. p-values (paired t-test) are shown above error bars.

The online version of this article includes the following figure supplement(s) for figure 4:

**Figure supplement 1.** Cardiac ischemia-reperfusion (IR) injury and glycative stress response in FEMALE WT vs.*Alkbh7*$^{-/-}$.

**Figure supplement 2.** Response to ex-vivo cardiac ischemia-reperfusion (IR) injury in OLD MALE WT vs.*Alkbh7*$^{-/-}$.

## Cardioprotection in Alkbh7$^{-/-}$ is not due to the mitochondrial unfolded protein response

We recently showed that activation of the mitochondrial unfolded protein response (UPR$^{mt}$) is sufficient to induce cardioprotection against IR injury (*Wang et al., 2019*). The genes encoding *LonP1* and *ClpP*, two mitochondrial proteases involved in UPR$^{mt}$ signaling (*Melber and Haynes, 2018*), are also located on mouse chromosome 17 adjacent to the *Alkbh7* gene. In addition, a recent proteomic study proposed a role for ALKBH7 in mitochondrial proteostasis (*Meng et al., 2019*) and our pull-down experiment identified several mitochondrial heat-shock proteins as potential ALKBH7 interactors (*Supplementary file 1*). Furthermore, the related protein ALKBH1 has been shown to partially localize to mitochondria, and its knock-down induces a UPR$^{mt}$ (*Wagner, 2019*). As such, we hypothesized constitutive UPR$^{mt}$ activation might underlie the cardioprotective effects of ALKBH7 ablation. However, western blotting (*Figure 5—figure supplement 1*) showed only small increases in LonP1 and ClpP protein in *Alkbh7$^{-/-}$* (the former non-significant), and a significant decrease in HSP60 (*Hspd1*) protein. We also did not find any UPR$^{mt}$ target proteins upregulated in our proteomics analysis (*Figure 2*). Overall, these observations suggest that modulation of the UPR$^{mt}$ is not a key mechanism by which ALKBH7 regulates necrosis.

## Cardioprotection in Alkbh7$^{-/-}$ is not via the mitochondrial permeability transition pore

A core component of the necrotic cell death machinery is the mitochondrial permeability transition (PT) pore, which is regulated by the cis/trans prolyl-isomerase cyclophilin D (CypD, *ppif*) (*Baines et al., 2005*). Parallels between CypD and ALKBH7 function have previously been speculated (*Wang et al., 2014*). In addition, although somewhat counter-intuitive, it has been shown that MGO can inhibit the PT pore (*Speer et al., 2003*), and our data suggest *Alkbh7$^{-/-}$* mice experience greater MGO stress (*Figures 2* and *3*). Thus, we hypothesized ALKBH7 may regulate the PT pore. However, an osmotic swelling PT pore assay in isolated cardiac mitochondria from WT and *Alkbh7$^{-/-}$* mice revealed only a slight blunting of pore opening in *Alkbh7$^{-/-}$* (*Figure 5A/B*). In addition, pore opening in both genotypes was inhibited by CypD inhibitor cyclosporin A, suggesting no differences in the underlying ability of CypD to regulate the pore. An isolated mitochondrial Ca$^{2+}$ handling assay (*Figure 5C–E*) showed a slight elevation in the amount of Ca$^{2+}$ required to trigger the pore in *Alkbh7$^{-/-}$*, and no difference in Ca$^{2+}$ uptake kinetics. Furthermore, blue-native gel analysis of ATP synthase multimers, which are postulated to contribute to the composition of the PT pore (*Giorgio et al., 2013*), showed no differences between *Alkbh7$^{-/-}$* and WT (*Figure 5—figure supplement 3*). Together these findings suggest that the mitochondrial PT pore is not a central mechanism by which ALKBH7 regulates necrosis.

## Cardioprotection in Alkbh7$^{-/-}$ requires glycolysis

Several paradigms of cardioprotection against IR injury have been linked to elevated glycolysis (*Nadtochiy et al., 2015*). Since *Alkbh7$^{-/-}$* mice exhibit elevated glycolysis, we decided to test the requirement for elevated glycolysis in the protected phenotype, by perfusing knockout hearts in the absence of glucose (i.e. fat as the only metabolic substrate). While no difference in baseline function was observed (thus indicating no overall defect in adapting to burning fat only), *Figure 6—figure supplement 1* shows that removal of glucose abrogated cardioprotection in *Alkbh7$^{-/-}$*. Contrary to observations with a rich substrate mix (*Figure 4*), infarct size was significantly greater in glucose-free-perfused *Alkbh7$^{-/-}$* hearts vs. WT. These data indicate that, as with several other modes of cardioprotection (e.g. ischemic preconditioning), glucose metabolism is a necessary component of the protection stemming from ALKBH7 loss.

## Cardioprotection in Alkbh7$^{-/-}$ requires GLO-1

Since we also showed that GLO-1 was elevated in response to ALKBH7 loss, we hypothesized this may also be an underlying component of the protected phenotype seen in the knockouts. To probe the requirement for elevated GLO-1 in cardioprotection, the GLO-1 inhibitor *S*-p-Bromobenzylglutathione cyclopentyl diester (SBB-GSH-CpE) was administered to hearts prior to ischemia. As *Figure 6* shows 1 µM SBB-GSH-CpE had no effect on WT hearts (c.f. *Figure 4*), but completely abrogated

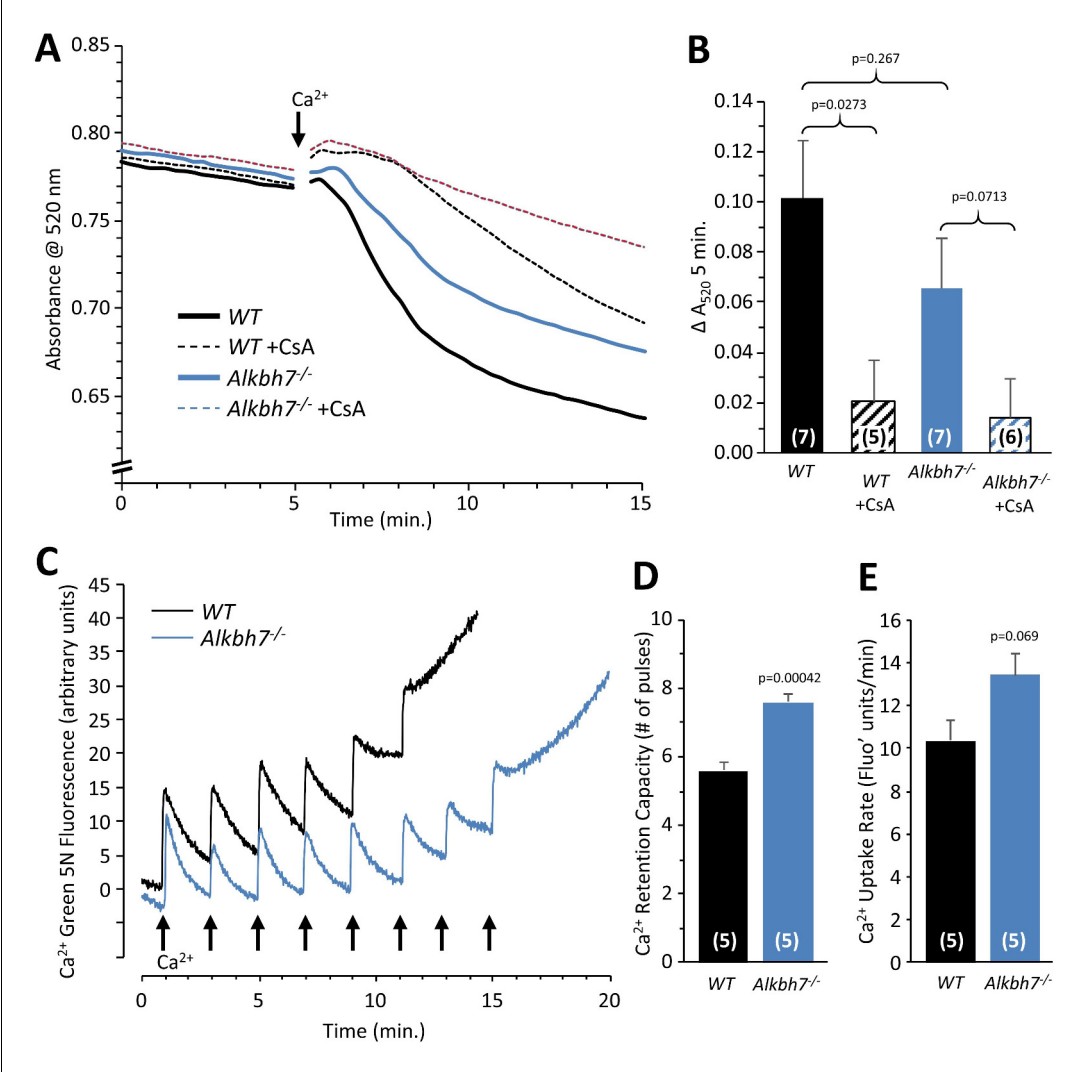

**Figure 5.** Mitochondrial PT pore and Ca$^{2+}$ handling in WT vs.*Alkbh7$^{-/-}$*. (**A**) Opening of the mitochondrial PT pore was assayed spectrophotometrically in isolated cardiac mitochondria from young male WT and *Alkbh7$^{-/-}$* mice. Average traces are shown, with addition of 100 μM Ca$^{2+}$ to initiate PT pore opening and swelling indicated by the arrow. Dotted lines indicate the presence of the PT pore inhibitor cyclosporin A (CsA). Error bars are omitted for clarity. (**B**) Quantitation of pore opening, as the change in swelling (absorbance at 520 nm) in 5 min. Data are means ± SE, N = 7, with significance between groups (unpaired t-test) shown above error bars. (**C**) Mitochondrial Ca$^{2+}$ handling assayed by Ca$^{2+}$ green-5N fluorescence. Isolated cardiac mitochondria from young male WT and *Alkbh7$^{-/-}$* mice were incubated with Ca$^{2+}$ green-5N to indicate extra-mitochondrial [Ca$^{2+}$]. Pulses of 10 μM Ca$^{2+}$ were added at ~2 min intervals as indicated by arrows. Representative traces are shown. (**D**) Quantitation of the number of Ca$^{2+}$ pulses tolerated by mitochondria before PT pore opening occurred (as indicated by a sharp upward deflection in the Ca$^{2+}$ green-5N trace). (**E**) Quantitation of the initial rate of mitochondrial Ca$^{2+}$ uptake, calculated from the downward slope in Ca$^{2+}$ green-5N fluorescence on the first 3 Ca$^{2+}$ pulses. Bar graphs in panels B/D/E show means ± SE, N = 5–7, with p-values (unpaired t-test) shown above error bars. In bar graphs, N for each group is shown in parentheses. The online version of this article includes the following figure supplement(s) for figure 5:

**Figure supplement 1.** Western blot detection of UPR$^{mt}$ mediators in WT vs.*Alkbh7$^{-/-}$*hearts.

**Figure supplement 2.** Cardioprotection against IR injury by dimethyl-L-2-hydroxyglutarate.

**Figure supplement 3.** Blue-native analysis of mitochondrial respiratory supercomplexes.

cardioprotection in *Alkbh7$^{-/-}$* hearts. Separate experiments to assay GLO-1 enzyme activity in SBB-GSH-CpE-treated hearts indicated this protocol resulted in 34 ± 8% GLO-1 inhibition (mean ± SD).

A significant depression of cardiac function was observed immediately upon SBB-GSH-CpE administration to *Alkbh7$^{-/-}$* hearts, with no effect in WT (*Figure 6A*). Due to its higher baseline level of MGO stress (*Figure 3D–F*), the *Alkbh7$^{-/-}$* heart is likely more dependent on GLO-1 activity and may therefore be hypersensitized to its inhibition. This finding suggests an important role for anti-

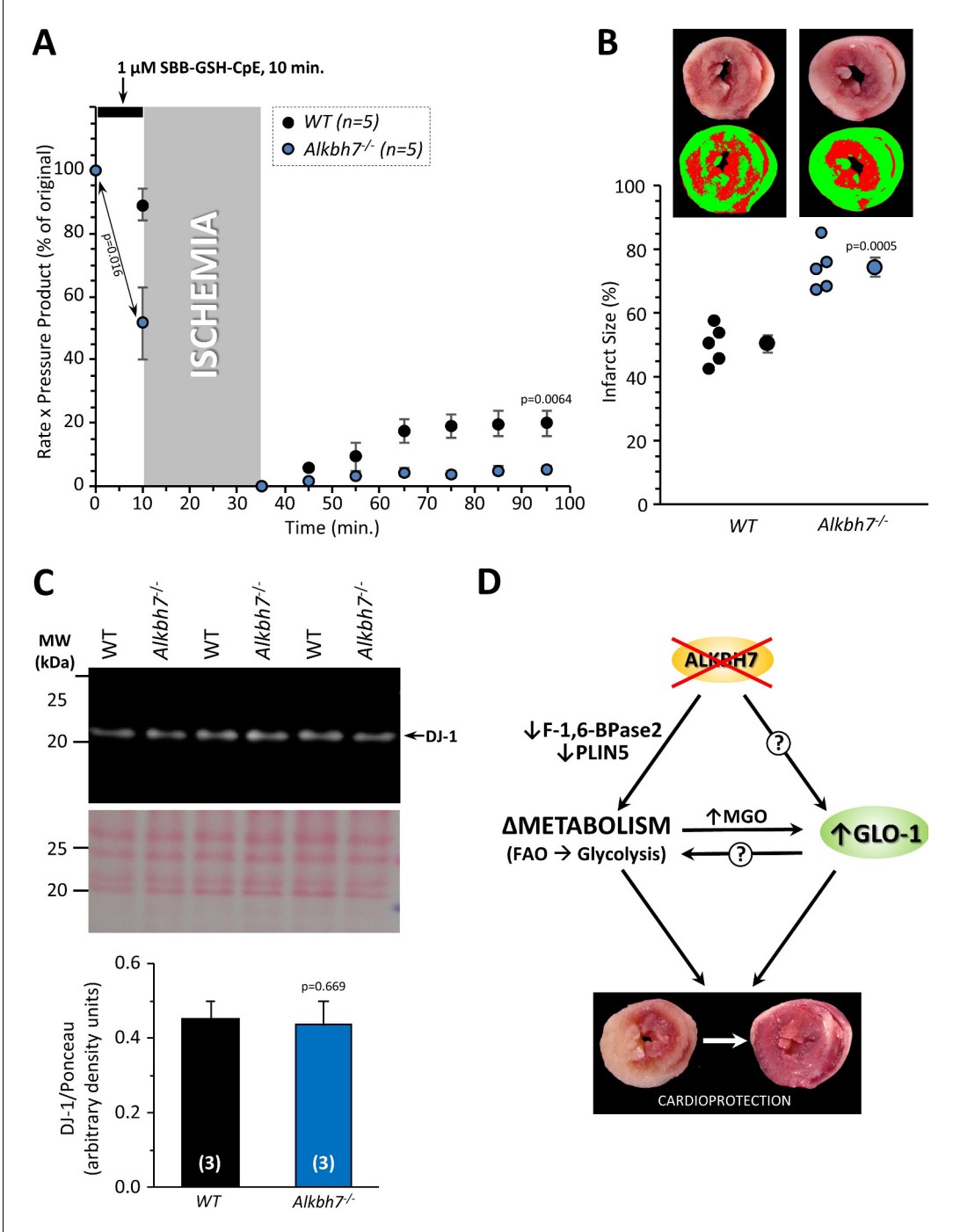

**Figure 6.** Blockade of cardioprotection in *Alkbh7⁻/⁻* by GLO-1 inhibition. Hearts from young male WT and *Alkbh7⁻/⁻* mice were Langendorff perfused and subjected to IR injury as in *Figure 4*, with delivery of 1 μM SBB-GSH-CpE for 10 min. prior to ischemia. (A) Cardiac function assessed by left ventricular balloon pressure transducer. Graph shows the product of heart rate multiplied by left ventricular developed pressure, as a percentage of the initial value. A significant drop in cardiac function was observed upon drug infusion in *Alkbh7⁻/⁻* only (see arrow and p-value). (B) Post IR staining with TTC for quantitation of myocardial infarct size. Representative TTC-stained heart slices are shown, with pseudo-colored mask images used for quantitation by planimetry (red = live tissue, green = infarct). Data are quantified below, with individual data points to shown N, and means ± SE. p-values (paired t-test) are shown above error bars. (C) Western blot showing abundance of DJ-1 in *Alkbh7⁻/⁻* and WT heart mitochondria. Ponceau stained membrane and quantitation are shown below. Bar graph shows means ± SE, N = 4, with p-values (paired t-test) shown above error bars. (D) Schematic showing proposed events that connect
*Figure 6 continued on next page*

*Figure 6 continued*

loss of ALKBH7 to cardioprotection. Via mechanisms that may include downregulation of Perilipin five and F-1,6-BPase 2, loss of ALKBH7 causes a shift in metabolism away from fatty acid oxidation (FAO) toward elevated glycolysis. GLO-1 is also upregulated, possibly in response to MGO elevation. Both elevated glycolysis and GLO-1 then protect against IR injury. The potential role of GLO-1 as a regulator of the metabolic shift toward glycolysis is also shown.

The online version of this article includes the following figure supplement(s) for figure 6:

**Figure supplement 1.** Response to ex-vivo cardiac ischemia-reperfusion (IR) injury in WT vs.*Alkbh7^{-/-}*, in the absence of glucose.

glycation enzymes such as GLO-1 in cardiac functional homeostasis. In this regard, the mitochondrial protein DJ-1 has been shown to function as a glyoxal detoxifying enzyme (*Lee et al., 2012*) and was also recently shown to confer cardioprotection (*Shimizu et al., 2016*; *Shimizu et al., 2020*). However, no differences in the levels of DJ-1 were observed in *Alkbh7^{-/-}* hearts (*Figure 6C*), suggesting that the signaling pathways engaged by ALKBH7 loss are somewhat specific to GLO-1 and may not engage other dialdehyde detoxification pathways.

## Discussion

Summarizing the current findings, a comprehensive analysis of the *Alkbh7^{-/-}* mouse heart suggests ALKBH7 is not a functional prolyl-hydroxylase that regulates mitochondrial activity, and that its role in necrosis involves rewiring of MGO metabolism. While our prolyl-hydroxylation proteomic findings were made at baseline, it may be argued that ALKBH7 could adopt a prolyl-hydroxylase function only under stress conditions. However, like all other α-KG dioxygenases, ALKBH7 uses oxygen as a substrate, so would be unlikely to engage additional hydroxylation substrates under oxygen-limited conditions such as ischemia.

We propose a scheme linking ALKBH7 loss to cardioprotection and glycative stress, as outlined in *Figure 6D*. Specifically, ALKBH7 loss triggers a mild defect in fat oxidation and an elevated rate of glycolysis, possibly via perilipin five and fructose-1,6-bisphosphatase 2 (proteomic data). The resulting glycative stress triggers rewiring of glyoxal metabolism, in particular GLO-1 induction, which then endows the added benefit of protection against IR injury. In addition, elevation of glycolysis per se may contribute to the cardioprotected phenotype (*Figure 6—figure supplement 1*; *Nadtochiy et al., 2015*). The relative position of GLO-1 as a downstream consequence of altered metabolism or as a driver of it, is currently unclear. In this regard, a recent systems genetic analysis identified *Glo1* as an important gene in the regulation of lipid metabolism (*Parker et al., 2019*), suggesting GLO-1 and metabolic fuel switching may be parallel responses to ALKBH7 loss. The exact mechanism by which ALKBH7, thought to be a mitochondrial protein, can communicate to GLO-1, a cytosolic protein, is unclear, although our finding that a portion of ALKBH7 protein is located in the cytosol raises the possibility that it is the cytosolic component of ALKBH7 which mediates these effects.

A surprising finding herein was that GLO-1 was upregulated in *Alkbh7^{-/-}* mice without concomitant upregulation of its companion enzyme GLO-2. The GLO-1/GLO-2 system (*Figure 2F*) typically recycles GSH, and as predicted *Alkbh7^{-/-}* mice exhibited depleted GSH levels. While such a finding might cast focus on oxidative stress as a phenotypic driver in *Alkbh7^{-/-}*, recent discoveries have implicated the GLO-1/GLO-2 system in epigenetic signaling. Specifically, the product of GLO-1, S-lactoyl-glutathione (SLG), has been shown to mediate the lactylation of lysine residues (*Gaffney et al., 2020*), including in histones (*Zhang et al., 2019*), which may represent a link between metabolism and gene regulation. As such, it is possible that SLG levels may be elevated in *Alkbh7^{-/-}*, which could drive epigenetic changes that underlie the phenotypes of the knockout. Unfortunately, attempts to quantify lysine-lactylation by western blotting were hampered by a cross-reactivity of the antibody with MGO adducts (pre-print at https://osf.io/sba8j).

The origins of the sexual dimorphism in necrosis and IR injury in *Alkbh7^{-/-}* mice remain unclear. In this regard, a recent study identified S-nitrosoglutathione reductase (GSNO-R) as a potential modulator of IR injury in male vs. female mice (*Casin et al., 2018*). Notably, GSNO-R can function as a formaldehyde dehydrogenase, and formaldehyde is a product of the DNA demethylation reaction

carried out by many ALKBHs. As such, it is possible that the role of ALKBH7 in necrosis may involve generation of formaldehyde, such that ALKBH7 deletion in males is protective by lowering the levels of this metabolite, whereas females have elevated GSNO-R levels so are already conditioned to lower formaldehyde levels.

The results herein may also provide insight to the complex biology of the diabetic heart. The incidence and progression of cardiac pathology such as heart failure is significantly worse in diabetes, and this is thought to be partly due to elevated glycative stress (*Ma et al., 2009*; *Kenny and Abel, 2019*). However, somewhat paradoxically the diabetic heart is relatively protected against acute IR injury (*Ravingerova et al., 2000*; *Nawata et al., 2002*; *Chen et al., 2006*). As such, it is interesting to speculate whether the mechanisms of ischemic tolerance seen in the male *Alkbh7⁻/⁻* heart, stemming from GLO-1 upregulation, may also apply to the diabetic heart. An additional ramification of the current results may be in the area of cancer biology, where there has been interest in the potential use of GLO-1 inhibitors to target metabolic vulnerabilities of cancer cells (*Gaffney et al., 2020*; *Rabbani et al., 2018*). The apparent cardiotoxic effects of SBB-GSH-CpE (*Figure 6*) suggest that caution may be required in the use of such drugs to ensure they do not elicit cardiac toxicity.

Although the precise biologic function and substrates of the ALKBH7 enzyme remain unknown, it is tempting to speculate that the native function of the enzyme may be in the detoxification of MGO or reactive intermediates prior to the formation of glycative adducts, such that ALKBH7 deletion drives the responses seen herein, to limit formation of MGO and its downstream intermediates. Recently, it was found that an R182Q substitution in ALKBH7 was positively selected for during the evolution of nectivorous bats (whose diet is mostly sugar) (*Gutiérrez-Guerrero et al., 2020*), thus furthering the notion that ALKBH7 is inherently linked to glucose metabolism, a major source of MGO. Overall, our findings highlight the importance of MGO homeostasis in the heart and suggest novel therapeutic targets for protection of tissues against IR injury. Further work is required to elucidate the signaling mechanisms that link the biochemical function of ALKBH7 to MGO metabolism.

# Materials and methods

## Animals and materials

*Alkbh7⁻/⁻* mice on a C57BL/6J background (*Jordan et al., 2017*; *Solberg et al., 2013*) were bred conventionally (WT x KO), PCR genotyped at weaning, and maintained according to the 'NIH Guide' (8th edition, 2011) on an IACUC approved protocol, with food and water available ad libitum. The colony has been back crossed into fresh C57BL/6J stocks for at least 20 generations. Since murine *Alkbh7⁻/⁻* phenotypes are only seen in males, primarily male mice were used (except where indicated), with littermate wild-type controls, at ages of 8–12 weeks (young) or 1.5 years (old). All procedures were performed following administration of heparin (250 units) and tribromoethanol anesthesia (200 mg/kg ip). All chemicals and other reagents were from Sigma-Aldrich (St. Louis MO) or VWR (Radnor PA), unless otherwise noted. For the synthesis of dimethyl-L-2-hydroxyglutarate, a drop of concentrated HCl was added to a solution of 0.5 g (S)-(+)−5-oxo-2-tetrahydrofurancarboxylic acid (Sigma-Aldrich #301469) in 3.84 ml dry MeOH. The reaction mixture was heated to reflux and stirred vigorously overnight, then quenched with saturated $NaHCO_3$, filtered and concentrated in vacuo. Crude material was purified via flash chromatography on silica (50% EtOAc in n-hexane) to obtain the desired product as a clear oil, as reported (*Winkler et al., 2013*).

## Isolated perfused hearts

Following heparin/anesthesia, the aorta was cannulated in-situ and the heart transferred to the perfusion apparatus, then retrograde perfused at 37°C with gassed (95% $O_2$/5% $CO_2$) Krebs-Henseleit buffer (KHB), as described (*Nadtochiy et al., 2015*). Left ventricular pressure was digitally recorded at 1 kHz via a transducer-linked water-filled balloon. Following 15 min equilibration, ischemia-reperfusion (IR) injury comprised 25 min global non-flow ischemia plus 60 min reperfusion. Hearts were then sliced and stained with triphenyltetrazoliumchloride for 20 min, fixed in 4% formalin for 24 hr., and slices digitally imaged for infarct size calculation by planimetry. KHB contained 5 mM glucose, 1.2 mM lactate, 0.5 mM pyruvate and 100 µM palmitate (conjugated 6:1 with fat-free bovine serum albumin) as metabolic substrates, unless indicated. The following experiments were conducted: (i) IR alone: WT and *Alkbh7⁻/⁻* hearts subjected to IR in three cohorts: young males, young females, old

males. (ii) GLO-1 inhibitor IR: Young male WT and *Alkbh7⁻/⁻* hearts were subjected to IR, with 1 µM SBB-GSH-CpE delivered for 10 min prior to ischemia. A small number of SBB-GSH-CpE-treated WT hearts were snap-frozen without ischemia, for measurement of GLO-1 activity. (iii) Glucose-free IR: Young male WT and *Alkbh7⁻/⁻* hearts were perfused with KHB containing palmitate-BSA alone (no glucose, lactate, pyruvate) and subjected to IR injury. (iv) Proteomics and steady-state metabolomics: Young male WT and *Alkbh7⁻/⁻* hearts were perfused for 15 min then snap-frozen and stored at −80° C until analysis. (v) Metabolic flux: Young male WT and *Alkbh7⁻/⁻* hearts were perfused with KHB. Following equilibration, glucose in KHB was replaced with 5 mM [U-$^{13}$C] glucose, and perfusion continued for 5 min, followed by snap-freezing and storage at −80°C until analysis. (vi) Dimethyl L-2-hydroxyglutarate plus IR: Young male WT hearts were subjected to IR, with 10 µM DM-L-2-HG delivered for 20 min prior to ischemia.

## Isolated mitochondrial experiments

Mouse heart mitochondria were isolated by differential centrifugation in sucrose-based media essentially as described (*Smith et al., 2018*; *Hoffman et al., 2007*). Following anesthesia, hearts were extirpated into ice-cold media comprising 300 mM sucrose, 20 mM Tris-HCl, 2 mM EGTA, pH 7.35 at 4°C. Tissue was chopped and washed twice to remove blood then homogenized in 4 ml media (IKA Tissumizer, 22,000 rpm). Homogenates were centrifuged at 800 x g, 5 min. Supernatants were centrifuged at 10,800 x g, 5 min, and pellets washed by a further two centrifugation steps with final resuspension in 30 µl. For Ca$^{2+}$ handling experiments (see below) the final spin utilized EGTA-free media. Protein was determined by the Lowry method (*Lowry et al., 1951*). Mitochondrial permeability transition (PT) pore opening, induced by 100 µM CaCl$_2$, was measured via spectrophotometric light scatter at 520 nm using a Beckman DU800 spectrophotometer, as described (*Brookes et al., 2000*). Mitochondria were incubated at 0.5 mg/ml in buffer comprising 120 mM KCl, 3 mM KH$_2$PO$_4$, 50 mM Tris, 5 mM succinate and 5 µM rotenone, pH 7.35 at 37 °C. Following CaCl$_2$ addition (100 µM), swelling was monitored for 20 min. In some incubations, cyclosporin A (5 µM) was added prior to CaCl$_2$.

Mitochondrial Ca$^{2+}$ handling was assayed using the fluorescent extra-mitochondrial dye Ca$^{2+}$-green-5N. Mitochondria were incubated at 0.25 mg/ml in buffer comprising 50 mM KCl, 150 mM sucrose, 2 mM KH$_2$PO$_4$, 20 mM Tris, 5 mM succinate, 5 µM rotenone and 500 nM Ca$^{2+}$-green-5N, pH 7.35 at 37 °C. Pulses of 10 µM CaCl$_2$ were added every 2 min., and fluorescence measured using an Agilent Cary Varian Eclipse spectrofluorimeter ($\lambda_{EX}$506 nm, $\lambda_{EM}$530 nm), as described (*Brookes et al., 2008*).

Liver mitochondria were isolated by differential centrifugation essentially as previously described (*Hoffman et al., 2007*). Livers were removed from anesthetized male WT and *Alkbh7⁻/⁻* mice and chopped into small pieces with double scissors in ice-cold liver mitochondria isolation medium (LMIM, 250 mM sucrose, 10 mM Tris hydrochloride, 1 mM EGTA, pH 7.4 at 4°C) and homogenized using a glass Dounce homogenizer. The homogenate was centrifuged at 1000 x *g* for 3 min and the supernatant decanted to a fresh tube, avoiding fat. This was followed by three rounds of centrifugation at 10,000 x *g*, 10 min, discarding the supernatant each time. The pellet was resuspended in 1 ml LMIM and protein quantified by the Lowry method (*Lowry et al., 1951*).

## Protein extraction and preparation for proteomics

Proteomic analysis was performed essentially as described (*Stoehr et al., 2016*). Male WT and *Alkbh7⁻/⁻* hearts for proteomic analysis were perfused in Langendorff mode as described above for 5 min, snap-frozen in liquid N$_2$ with Wollenberger tongs, then ground to powder and stored in two portions at −80°C. One half of each heart was shipped on dry ice from Rochester NY to NHLBI (Bethesda MD) for proteomic analysis. Frozen heart samples were homogenized in 280 mM sucrose, 10 mM HEPES, 1 mM EGTA, and 1% (w/v) laurylmaltoside supplemented with 1X protease inhibitors (Millipore Sigma #4693159001) and 1X phosphatase inhibitors (Millipore Sigma # 4906837001) in a Precellys 24 with Cryolys (Bertin technologies). Protein was determined using a Bradford assay (Sigma #B6916), and 100 µg protein was brought to 100 µl final volume with 100 mM triethylammonium bicarbonate (TEAB). Protein was reduced with 10 mM dithiothreitol at 55°C for 60 min rocking at 650 rpm, then alkylated with 18 mM iodoacetamide for 60 min protected from light. Protein was precipitated overnight with six volumes acetone at −20°C, then resuspended 100 mM

triethylammonium bicarbonate (TEAB) and sonicated briefly in a chilled water bath sonicator. Protein was digested with 50 µg Trypsin (Promega #V5111) overnight at 37°C with shaking at 650 rpm. Tryptic peptides were tagged with Tandem Mass Tag (TMT) labeling reagents (Thermo Fisher #90110 and #A37724) according to the manufacturer's instructions. Labeled peptides were then lyophilized, resuspended in 50 mM ammonium bicarbonate, and layered over resin (G-Biosciences #GBS10-800) to remove residual detergent, according to manufacturer instructions. Eluted labeled tryptic peptides were lyophilized, resuspended in 0.1% (v/v) formic acid (FA), and desalted using Hydrophilic-Lipophilic-Balanced (HLB) columns (Waters #186000383) according to manufacturer instructions. Eluted peptides were lyophilized and resuspended for off-line fractionation.

## HPLC fractionation and LC-MS analysis

Dried and labeled tryptic peptides were reconstituted with basic reverse-phase liquid chromatographic (bRPLC) buffer A (10 mM TEAB, pH 8.0) and separated using a C18 column (Xbridge 130 Å, 3.5 µm, 4.6 mm x 150 mm, Waters) on a 1200 series HPLC (Agilent). The linear gradient comprised 5–40% solvent B (10 mM TEAB, acetonitrile, pH 8.0) over 96 min, with fractions collected every minute. The 96 fractions were later combined manually to 17 fractions and lyophilized.

Protein identification by LC-MS/MS employed an Orbitrap Fusion Lumos Tribid mass spectrometer (Thermo Scientific) interfaced with an Ultimate 3000 Nano-HPLC apparatus (Thermo Scientific). Peptides were fractionated by EASY-Spray PepMAP RPLC C18 column (2 µm, 100 Å, 75 µm x 50 cm) using a 120 min linear gradient of 5–35% acetonitrile in 0.1% FA at 300 nl/min flow rate. The instrument was operated in data-dependent acquisition mode (DDA) using fourier transform (FT) mass analyzer for one survey MS scan. This was done on selected precursor ions followed by top 3 s data-dependent higher-energy collision (HCD)-MS/MS scans for precursor peptides with 2–7 charged ions above a threshold ion count of 10,000 with normalized collision energy of 37%. Survey scans of peptide precursors from 300 to 2000 m/z were performed at 120 k resolution and MS/MS scans were acquired at 50,000 resolution with a m/z range 100–2000.

## Protein identification and analysis

All MS and MS/MS raw spectra of TMT experiments were processed and searched using Sequest HT and Mascot algorithms within Proteome Discoverer 2.2 software (PD2.2, Thermo Scientific). Precursor mass tolerance was set at 12 ppm, fragment ion mass tolerance to 0.05 Da, trypsin enzyme with 2 mis cleavages. Carbamidomethylation of cysteine was set as a fixed modification; and TMT 6-plex (lysine), TMT 6-plex (N-term), deamidation of glutamine and asparagine, oxidation of proline and methionine were set as variable modifications. The mouse sequence database from Swiss-prot was used for database search. Identified peptides were filtered for maximum 1% false discovery rate (FDR) using the Percolator algorithm in PD 2.2 along with additional peptide confidence set to high. The final lists of protein identification and quantitation were filtered by PD 2.2 with at least two unique peptides per protein identified with medium confidence.

The method overall detected 49,427 peptides representing 5642 proteins. Filtering for proteins with more than two peptides identified in either search engine (Mascot or Sequest) yielded 3737 proteins with an average 29.13% sequence coverage (95% confidence interval 28.49–29.76%). Filtering the total peptide set for P-OH containing peptides yielded 625 peptides, with 451 of these having a corresponding abundance value for the parent protein, originating from a total of 238 individual proteins. The abundance of each P-OH containing peptide was normalized to abundance of its parent protein, to determine relative hydroxylation levels between WT and *Alkbh7*$^{-/-}$ paired samples.

## Immunoprecipitation to identify ALKBH7 binding partners

The coding region for human ALKBH7 was and cloned into pcDNA3.1 (Invitrogen) for expression as C-terminal 3xFLAG tag fusion protein. Transient transfection and cellular extract production were performed as previously described (*Fu et al., 2010*). Negative control cells were transfected with vector only. Briefly, $2.5 \times 106$ HEK 293 T cells were transiently transfected by calcium phosphate DNA precipitation with 20 µg of plasmid DNA, followed by preparation of the lysate by hypotonic freeze-thaw lysis at 48 hr. post-transfection. Whole-cell extract was rotated with 10 µl of FLAG M2 antibody resin (Sigma) for 2 hr. at 4°C in lysis buffer (150 mM NaCl, 20 mM HEPES, 2 mM MgCl2,

0.2 mM EGTA, 10% (v/v) glycerol, 1 mM dithiothreitol, 0.1 mM phenylmethylsulfonyl fluoride, 0.1% (v/v) NP-40, pH 7.9). Resin was washed extensively using the same buffer, and bound proteins eluted with two sequential volumes of wash buffer containing 100 µg/ml of 3 × FLAG peptide (Sigma). Frozen stocks of the HEK293 cell line were obtained at least annually from ATCC (Manassas VA). Cell line identity was not independently authenticated by us, but cells were confirmed to be free of mycoplasma contamination.

Protein identification was performed by the MIT Center for Cancer Research Biopolymers Laboratory (https://ki.mit.edu/sbc/biopolymers). Gel slices of protein bands were excised, reduced, alkylated, and digested in solution with trypsin, followed by purification and desalting of peptides on analytical C18 column tips. Peptide samples were analyzed by chromatography on an Agilent model 1100 Nanoflow high-pressure liquid chromatography (HPLC) system coupled by electrospray ionization to a Thermo LTQ ion-trap mass spectrometer. Protein identification through tandem mass spectrum correlation was performed using SEQUEST. Spectra had to match full tryptic peptides of at least seven amino acids, have a normalized difference in cross-correlation scores (ΔCn) of at least 0.1, and have minimum cross-correlation scores (Xcorr) of 1.8 for singly charged, 2.5 for doubly charged, and 3.5 for triply charged spectra with at least 50% ion coverage. Proteins from the vector-only control condition were eliminated from the mass spectrometric analysis.

## Metabolomics

Male WT and $Alkbh7^{-/-}$ hearts for metabolomics analysis were perfused in Langendorff mode as described above for 5 min, snap-frozen in liquid $N_2$ with Wollenberger tongs, then ground to powder and stored in two portions at −80°C. Heart powder was serially extracted in 80% aqueous methanol, extracts evaporated to dryness under $N_2$ and resuspended in 50% aqueous methanol. Liquid chromatography-tandem mass spectrometry (LC-MS/MS) analysis was performed by resolving metabolites on a Synergi Fusion RP C18 column (Phenomenex, Torrance, CA) with an acetonitrile elution ramp. Metabolites were identified by retention times and by single reaction monitoring (SRM) on a Thermo Quantum TSQ triple-quadrupole mass spectrometer (Thermo Scientific, Waltham, MA) as previously described (*Nadtochiy et al., 2015*; *Zhang et al., 2018*).

Metabolite identification used a custom SRM library for which fragmentation patterns including confirming ions at different collision energies were empirically determined from a library of purchased chemical standards. Data were analyzed using XCalibur Qual Browser (Thermo Scientific), with relative metabolite content being normalized to the sum of all metabolites in each sample run. Eight pairs of samples were prepared and analyzed in April 2017 in a core facility setting, yielding data for 61 metabolites. A further nine pairs of samples were prepared and analyzed in April 2019 in the senior author's laboratory, yielding additional data for 71 metabolites. Overall the analysis included 90 metabolites in total with 43 common between both data sets. As such the number of biological replicates varied between 8 and 17 depending on the metabolite in question. To process the metabolomic data set, outliers were flagged where the group-wise (WT or $Alkbh7^{-/-}$) data for a given metabolite exhibited a greater than 25% standard error, and individual values outside the 95% confidence intervals were removed. Missing values were imputed as weighted medians (*Aittokallio, 2010*) ,only in situations where more than 75% of original values were still present. Of a potential 2286 total data points, 21 outliers and 32 missing values were imputed, representing 2.3% of the total data set.

For the measurement of glycolytic flux, following 20 min of stable normoxic perfusion, $^{12}$C glucose in KH buffer was replaced with [U-$^{13}$C] glucose, and hearts perfused for a further 5 min. Hearts were then freeze-clamped and processed similar to steady-state metabolomics, with a custom SRM library used to detect isotopologues of common metabolites. Fractional saturation of selected metabolites with $^{13}$C label was determined, with correction for natural $^{13}$C abundance, as described previously (*Nadtochiy et al., 2015*; *Zhang et al., 2018*).

## Western blotting

Hearts from male WT and $Alkbh7^{-/-}$ mice were fractionated by differential centrifugation as previously described (*Nadtochiy et al., 2018*). Protein content was determined by the Folin-Phenol (Lowry) assay (*Lowry et al., 1951*). Non-mitochondrial samples were diluted two-fold in SDS-PAGE sample loading buffer and incubated at 100°C for 5 min. Mitochondrial samples were diluted two-

fold in sample loading buffer containing five times the standard concentration of SDS and incubated at 25°C for 30 min. Samples were separated by SDS-PAGE (12.5% or 15% gels) and transferred to 0.2 μm nitrocellulose membranes and probed with antibodies as recommended by manufacturer's protocols. Antibodies used include anti-ALKBH7 (#A2331, Abclonal, Woburn MA), anti-HSPD1 (#AP2859b Abgent, San Diego, CA), anti-LONP1 (#AP19551c Abgent), anti-CLPP (#PA5-79051 Thermo-Fisher, Waltham MA), anti-methylglyoxal (#ab243074 Abcam, Cambridge, MA), anti-GLO1 (#ab137098 Abcam), anti-HADHSC (#sc-376525 Santa Cruz Biotech', Dallas TX), Anti DJ-1 (#2134, Cell Signaling Technology, Danvers, MA), and anti ANT1 (#ab110322, Abcam ). Detection employed horseradish peroxidase-linked secondary antibodies with chemiluminescent detection (KwikQuant, Kindle Bioscience, Greenwich, CT). Sample loading was normalized to Ponceau S staining of membranes immediately after transfer. For most blots frozen samples were used following appropriate fractionation, but for MGO adduct blots we found the signal decayed over time with sample storage at minus 80°C, so freshly harvested samples were used.

## Blue-native electrophoresis

Mitochondrial respiratory supercomplexes were extracted and analyzed essentially as described by *Beutner et al., 2017*. Briefly, mitochondria (0.5 mg/ml) were incubated for 5 min in respiration buffer comprising 120 mM KCl, 10 mM HEPES, 1 mM EGTA, 5 mM $KH_2PO_4$, 5 mM $MgCl_2$, pH 7.3 at 37 °C. After centrifugation (14,000 x *g*, 10 min) pellets were suspended in 25 μl of buffer comprising 50 mM NaCl, 40 mM imidazole, 2 mM aminocaproic acid, 1 mM EDTA, 5.7% (w/v) digitonin, pH 7 at 4 °C, and incubated on ice for 20 min. Samples were then centrifuged (14,000 x *g*, 10 min), supernatants mixed 1:1 with loading buffer (50 mM aminocaproic acid, 5% (w/v) Coomassie Blue-G), followed by resolution on 5–8% gradient blue-native gels. For the complex V in-gel assay, the gel was incubated for 2 hr. in buffer comprising 35 mM Tris, 270 mM glycine, pH 8.3 at 25°C. White precipitate complex V activity bands were visualized by adding 135 mM $MgSO_4$, 6.5 mM $Pb(NO_3)_2$, and 7.8 mM ATP to the buffer. The reaction was stopped by adding 50% methanol and gel imaged.

## Cardiomyocyte isolation and seahorse respirometry

$Ca^{2+}$ tolerant primary adult cardiomyocytes were isolated from male WT and *Alkbh7$^{-/-}$* mouse hearts by collagenase digestion as previously described (*Smith et al., 2018*; *Zhang et al., 2018*). yielding ~800,000 rod-shaped cells with >80% viability by Trypan blue assay. The final cell pellet was divided into two portions, each of which was suspended in 1 ml cardiomyocyte incubation buffer (glucose-free DMEM supplemented with 4 mM L-glutamine, 10 mM HEPES, 100 μM sodium pyruvate, 5 mM D-glucose, 500 μM L-carnitine hydrochloride, and 100 μM oleate conjugated 6:1 to fat-free bovine serum albumin (BSA,)pH 7.4 at 37°C). From this suspension, cells were seeded at 2000/ well on laminin-coated Seahorse XF96 V3-PS plates (Agilent, Santa Clara, CA) and incubated for 1 hr in a 37°C humidified incubator. Cardiomyocyte incubation buffer was replaced with unbuffered DMEM (pH 7.4) containing 4 mM L-glutamine, 100 μM sodium pyruvate, 10 mM 2-deoxy-D-glucose, 500 μM L-carnitine hydrochloride and 100 μM oleate conjugated to BSA. The plate was incubated for 30 min at 37°C, and then oxygen consumption rate (OCR) was measured with a Seahorse XF96 extracellular flux analyzer (Agilent) at baseline and following sequential injections of 1 μM FCCP + 1 μg/mL oligomycin, 5 μM etomoxir and 1 μM antimycin A + 5 μM rotenone.

## Enzyme assays

Enzyme activities were determined in isolated mitochondria and cytosol from hearts and livers of WT and *Alkbh7$^{-/-}$* mice, as indicated. Mitochondria were freeze/thawed 3x. Complex I was measured spectrophotometrically at 340 nm as the rotenone-sensitive, coenzyme $Q_1$-linked oxidation of NADH, as previously reported (*Porter et al., 2014*). Cardiac or liver mitochondria were incubated in potassium phosphate buffer (pH 7.2) at 37°C containing 2.5 mg/ml BSA, 1 mM KCN, 75 μM NADH. NADH oxidation was followed at 340 nm (ε = 6180 $M^{-1}cm^{-1}$) for 5 min after addition of 100 μM coenzyme $Q_1$. At the end of each run, 10 μM rotenone was added and the rotenone-insensitive rate subtracted.

Complex II was measured spectrophotometrically at 60 nm as the rate of succinate-driven, thenoyltrifluoroacetone (TTFA)-sensitive, co-enzyme $Q_2$-linked reduction of dichlorophenolindophenol (DCPIP) as previously reported (*Wang et al., 2017*). Cardiac and liver mitochondria were incubated

in potassium phosphate buffer (pH 7.4) at 37°C containing 120 μM DCPIP, 1 mM KCN, 10 μM rotenone, and 50 μM co-enzyme Q$_2$. The rate of DCPIP reduction was followed at 600 nM ($\varepsilon$ = 21 mM$^{-1}$cm$^{-1}$) for 5 min after addition of 5 μM succinate. At the end of each run, 1 mM TTFA was added and the TTFA-insensitive rate subtracted.

α-Ketoglutarate dehydrogenase was measured spectrophotometrically at 340 nm as the α-ketoglutarate-dependent, 2-Keto-3-methyl-valerate (KMV) sensitive reduction of NAD$^+$, as described (*Chouchani et al., 2010*). Cardiac and liver mitochondria were incubated in assay buffer comprising 35 mM KH$_2$PO$_4$, 5 mM MgCl$_2$, 0.5 mM EDTA, 0.05 % v/v Triton X-100, 500 μM NAD$^+$, 200 μM thiamine pyrophosphate, 40 μM reduced Coenzyme-A, 2 mM KCN, 25 μM rotenone, pH 7.25 at 37°C. The reaction was initiated by addition of 2 mM α-ketoglutarate and the rate of NAD$^+$ reduction measured at 340 nm ($\varepsilon$ = 6220 M$^{-1}$cm$^{-1}$) for 5 min. At the end of each run, 25 mM KMV was added and the KMV-insensitive rate subtracted.

Citrate synthase was measured spectrophotometrically at 412 nm as the oxaloacetate and Acetyl CoA-linked production of 2-nitro-5-thiobenzoate (TNB) from 5,5'-dithiobis-(2-nitrobenzoic acid, DTNB) (*Trounce et al., 1996*). Cardiac and liver mitochondria were incubated in assay buffer (100 mM Tris, 0.1 % v/v Tritox X-100, 100 μM acetyl CoA, 200 μM DTNB, pH 8.0 at 37°C), the reaction initiated by addition of 200 μM oxaloacetate, and the initial linear (pre-plateau) rate of TNB formation measured at 412 nm ($\varepsilon$ = 13.6 mM$^{-1}$cm$^{-1}$).

The activity of short chain and long chain specific isoforms 3-HydroxyacylCoA dehydrogenase (HADH) was measured spectrophotometrically at 340 nm as the corresponding 3-oxoacyl CoA-linked oxidation of NADH, following literature procedure (*Wanders et al., 1990*; *Weinberger et al., 1995*). Cardiac and liver mitochondria were incubated in potassium phosphate buffer (pH 6.3) at 37°C containing 100 μM NADH, 100 μM dithiothreitol and 0.1 % w/v Triton X-100. The reaction was initiated by addition of 50 μM 3-ketopalmitoyl CoA (for long chain HADH) or 50 μM acetoacetyl CoA (for short chain HADH) and the initial liner rate of NADH oxidation followed at 340 nm ($\varepsilon$ = 6.22 mM$^{-1}$cm$^{-1}$).

Glyoxalase I (GLO-1) activity was measured spectrophotometrically at 240 nm as the rate of formation of *S*-D-lactoylglutathione (SLG) from the hemithioacetal adduct pre-formed in situ by incubation of methylglyoxal and glutathione, as reported (*Arai et al., 2014*). 2 mM glutathione and 2 mM methylglyoxal were incubated in 50 mM sodium phosphate buffer (pH 6.6) at 37°C for 10 min. Cardiac or liver cytosolic extracts were then added, and the initial linear rate of SLG formation followed at 240 nm ($\varepsilon$ = 2.86 mM$^{-1}$cm$^{-1}$) for 5 min.

Glyoxalase II (GLO-2) activity was measured spectrophotometrically at 240 nm as the rate of hydrolysis of SLG, as reported (*Arai et al., 2014*). 30 μM SLG was incubated in 50 mM Tris HCl buffer (pH 7.4) at 37°C. Cardiac or liver cytosolic extracts were added to the cuvette and the initial linear rate of SLG hydrolysis followed at 240 nm ($\varepsilon$ = 3.10 mM$^{-1}$cm$^{-1}$) for 5 min.

## Statistics

For all experiments, a single N (biological replicate) was considered to be the material arising from a single animal. N ranged from 3 to 17 depending on experiment, and is indicated in each figure or legend. Statistical significance was assessed by ANOVA with post-hoc Student's t-test. Where appropriate (comparisons between littermate paired samples), paired t-tests were used. Samples from WT and *Alkbh7*$^{-/-}$ were run in random order, and whenever possible experiments were performed in a blinded manner with the experimenter agnostic to the identity of the samples.

## Acknowledgements

Work in the laboratory of PSB is funded by a grant from the US National Institutes of Health (R01-HL071158). CAK is funded by a post-doctoral fellowship from the American Heart Association (#19POST34380212). LK and EM are funded by the NHLBI-NIH Intramural Research Program (ZO1-HL002066). DF was funded by a Rochester Aging Research Center Pilot Grant. We acknowledge S Patel and M Gucek in the NHLBI Proteomics Core Facility for collaborating on the proteomics in murine hearts. We thank Rudi Fasan (Rochester) for help with synthesis of dimethyl-L-2HG.

# Additional information

## Funding

| Funder | Grant reference number | Author |
|---|---|---|
| National Institutes of Health | R01-HL071158 | Paul S Brookes |
| American Heart Association | #19POST34380212 | Chaitanya A Kulkarni |
| NIH Office of the Director | ZO1-HL002066 | Leslie Kennedy Elizabeth Murphy |
| Medical Center, University of Rochester | Rochester Aging Research Center Pilot grant | Dragony Fu |

The funders had no role in study design, data collection and interpretation, or the decision to submit the work for publication.

## Author contributions

Chaitanya A Kulkarni, Data curation, Formal analysis, Investigation, Writing - review and editing; Sergiy M Nadtochiy, Leslie Kennedy, Data curation, Investigation, Writing - review and editing; Jimmy Zhang, Sophea Chhim, Investigation; Hanan Alwaseem, Resources, Investigation; Elizabeth Murphy, Investigation, Writing - review and editing; Dragony Fu, Conceptualization, Formal analysis, Investigation, Project administration, Writing - review and editing; Paul S Brookes, Conceptualization, Resources, Data curation, Formal analysis, Supervision, Funding acquisition, Investigation, Writing - original draft, Project administration, Writing - review and editing

## Author ORCIDs

Chaitanya A Kulkarni (iD) http://orcid.org/0000-0002-6836-0518
Dragony Fu (iD) http://orcid.org/0000-0002-8725-8658
Paul S Brookes (iD) https://orcid.org/0000-0002-8639-8413

## Ethics

Animal experimentation: All animal work was conducted according to the "NIH Guide" (8th edition, 2011). Animals were housed in an AAALAC accredited facility with food and water available ad libitum. All procedures were performed under tribromoethanol anesthesia. All animal work was approved by the University of Rochester Committee on Animal Resources (UCAR protocol # 2007-087).

## Decision letter and Author response

Decision letter https://doi.org/10.7554/eLife.58573.sa1
Author response https://doi.org/10.7554/eLife.58573.sa2

# Additional files

## Supplementary files

• Source data 1. Source data for *Figure 1* and Supplements.

• Source data 2. Source data for *Figure 2* and Supplements.

• Source data 3. Source data for *Figure 3* and Supplements.

• Source data 4. Source data for *Figure 4* and Supplements.

• Source data 5. Source data for *Figure 5* and Supplements.

• Source data 6. Source data for *Figure 6* and Supplements.

• Supplementary file 1. Proteins identified as interacting with ALKBH7 by immunoprecipitation under control or MMS-treated conditions. Cells transfected with FLAG-tagged ALKBH7 were treated with MMS as per the methods, the anti-FLAG beads used to immunoprecipitate ALKBH7-interacting proteins. Following SDS-PAGE separation (see *Figure 2—figure supplement 1*), excised bands were

identified by mass spectrometry of trypsin digests. Proteins are listed by name and accession, with the number of unique peptides identified under each condition indicated in the appropriate columns.

- Transparent reporting form

## Data availability

The complete original data set used to generate all figures is attached as a Microsoft Excel file, with the submitted files. A DOI has been reserved at the data sharing site FigShare (https://doi.org/10.6084/m9.figshare.12200273) and the file has been uploaded there.

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
