## [Decision Letter]

**Acceptance summary:**

Using hearts from ALKBH7^-/-^ mice and the corresponding WT littermates as controls, the authors provide novel mechanistic insights into the biological role of ALKBH7, a poorly characterized mitochondrial α-Ketoglutarate dependent dioxygenase for which no known substrates have been identified.

**Decision letter after peer review:**

Thank you for submitting your article "ALKBH7 mediates necrosis via rewiring of glyoxal metabolism" for consideration by *eLife*. Your article has been reviewed by three peer reviewers, and the evaluation has been overseen by a Reviewing Editor and Matthias Barton, MD as the Senior Editor. The following individuals involved in review of your submission have agreed to reveal their identity: James Galligan (Reviewer #1); John Calvert (Reviewer #2).

The reviewers have discussed the reviews with one another and the Reviewing Editor has drafted this decision to help you prepare a revised submission.

Summary:

In the current study, the authors sought to examine the role of ALKBH7 in the control of necrosis following the onset of myocardial ischemia-reperfusion injury. Using a multi-omics approach, the authors found that ALKBH7 does not seem to function as a prolyl-hydroxlase or mediate changes in the heart via protein-protein interactions. Rather, ALKBH7 deficient hearts have elevated levels of glycative stress and an upregulation in the expression of the anti-glycative stress protein glyoxylase-1 (GLO-1). The latter is shown to mediate the cardioprotection found in the hearts of ALKBH7 deficient mice. Overall, the study provides a comprehensive assessment of how ALKBH7 deficiency alters glyoxal metabolism in the heart and how this can impart protection against ischemic injury. The experimental approach is appropriate and the results are presented in a clear manner. Remaining issues are related to supporting the conclusions that ALKBH7 elicits hormetic signaling to upregulate GLO-1.

Essential revisions:

– The authors state "it is tempting to speculate that the native function of the enzyme may be in the repair of glycative adducts, such that its deletion drives the responses seen herein to limit formation of such adducts" but this is not very likely, given the structure of these adducts, particularly MG-H1. This hydroimizalone is highly stable and not likely to undergo any type of enzymatic removal. The removal of the intermediate, however, is more likely (akin to the proposed mechanism of action of DJ-1). Further, the removal of the adduct, itself, is unlikely to contribute substantially to the elevated concentrations of methylglyoxal observed. Thus, the more likely scenario is that ALKBH7 may detoxify methylglyoxal. Given that the substrates for this enzyme are "poorly characterized" this may be the mechanism by which ALKBH7 knockout results in increased methylglyoxal, thus leading to the increased GLO-1 protein expression. This notion should at least be discussed in the manuscript.

– How were non-specific *Alkbh7* interacting proteins controlled for in the FLAG-pulldown experiments? Gel images should be included in the supplement.

– Given the suggestion that *Alkbh7* may regulate the glyoxalase pathway (or act on methylglyoxal, directly), it is important to quantify the levels of methylglyoxal and SLG. The western blotting data is not overtly convincing and the authors clearly have the MS capability to carry out a more elegant quantification of glyoxalase function/activity. This can also be done with the 13C flux experiment. Do the authors see elevations in 13C-methylglyoxal and/or 13C-SLG? Lastly, the reduction in GSH and the combination of increased GLO-1 and (modestly) reduced GLO-2 certainly suggests that suggests that SLG would be increased, rather than methylglyoxal, and the effects observed may be due to increased SLG, rather than increased methylglyoxal.

– In line with the point addressed above, is it possible that the female mice do not display these changes in methylglyoxal metabolism and this is the reason for the sexual dimorphism? Inclusion of the female mice is a bit confusing but perhaps this is a useful "control" for the hormetic effects of methylglyoxal?

– Figure 6: SBB-GSH-CpE is simply a modified GSH and thus very likely to act on multiple enzymes that act on GSH and/or GSH detoxification products. While this is a commercial inhibitor "specific for GLO-1" and no characterization of the off-target effects of this compound is required, the levels of methylglyoxal, at minimum, must be measured here to demonstrate that the inhibitor is, in fact, elevating the levels of methylglyoxal.

– Figure 6: are methylglyoxal-derived modifications elevated following SBB-GSH-CpE treatment?

– The authors appear to have quantified α-KG in their metabolomics supplement, with no statistical changes being noted. Given the dependence of *Alkbh7* on α-KG, the authors should state that this was measured, indicating no significant changes. Without demonstrating elevations in methylglyoxal and/or SLG, it is hard to conclusively argue the hormetic effects of methylglyoxal in this context.

– The evidence for the induction of hormetic signaling in the absence of ALKBH7 is not fully supported. Specifically, the link directly tying ALKBH7 to GLO-1 is missing. The authors suggest that Nrf2 could be contributing. This would indicate that elevated levels of oxidative stress could be in play. Additionally, the authors indicate that SLG could be a link. In both cases, evidence to support either potential mechanism is not provided. Some insights are needed to support the hormetic-signaling hypothesis.

– The authors clearly show that removing glucose from the perfusate attenuates the protective effects of ALKBH7 in the setting of myocardial ischemia injury. From this data, the authors suggest that this supports a role for elevated glyoxal metabolism. However, the MGO levels should still be elevated in the hearts and more importantly GLO-1 levels should still be elevated. It seems to reason that this data would indicate a deficit in fat utilization.

– The authors provide evidence that the deficiency of ALKBH7 does not provide protection in the female heart and that with age male ALKBH7 mice lose protection against ischemic injury. It would be nice to know if the levels of MGO or GLO-1 are altered in either instance.

– The authors provide compelling evidence for minimal changes in hydroxyproline levels in ALKBH7 mice at baseline. While this is indicative of basal conditions, there is not any evidence related to the activity of ALKBH7 in the setting of myocardial ischemia. If ALKBH7 is a stress-induced protein, it is possible that changes in hydroxyproline-modified proteins could become evident under these conditions.

---

## [Author Response]

Essential revisions:– The authors state "it is tempting to speculate that the native function of the enzyme may be in the repair of glycative adducts, such that its deletion drives the responses seen herein to limit formation of such adducts" but this is not very likely, given the structure of these adducts, particularly MG-H1. This hydroimizalone is highly stable and not likely to undergo any type of enzymatic removal. The removal of the intermediate, however, is more likely (akin to the proposed mechanism of action of DJ-1). Further, the removal of the adduct, itself, is unlikely to contribute substantially to the elevated concentrations of methylglyoxal observed. Thus, the more likely scenario is that ALKBH7 may detoxify methylglyoxal. Given that the substrates for this enzyme are "poorly characterized" this may be the mechanism by which ALKBH7 knockout results in increased methylglyoxal, thus leading to the increased GLO-1 protein expression. This notion should at least be discussed in the manuscript.

Thank-you for the additional insight regarding whether the enzyme would be anticipated to act on glycation adducts. We agree that a more likely interpretation is that the enzyme may work directly on MGO or a reactive intermediate. We have now edited the text (Discussion) to reflect this.

– How were non-specific Alkbh7 interacting proteins controlled for in the FLAG-pulldown experiments? Gel images should be included in the supplement.

Parallel to the FLAG-tagged ALKBH7 cells, pull-down was performed from vector-only expressing cells. A representative gel image is now shown in the supplement (Figure 2—figure supplement 1), and the Materials and methods section has been updated to include this detail (subsection “Immunoprecipitation to identify ALKBH7 binding partners”).

– Given the suggestion that Alkbh7 may regulate the glyoxalase pathway (or act on methylglyoxal, directly), it is important to quantify the levels of methylglyoxal and SLG. The western blotting data is not overtly convincing and the authors clearly have the MS capability to carry out a more elegant quantification of glyoxalase function/activity. This can also be done with the 13C flux experiment. Do the authors see elevations in 13C-methylglyoxal and/or 13C-SLG? Lastly, the reduction in GSH and the combination of increased GLO-1 and (modestly) reduced GLO-2 certainly suggests that suggests that SLG would be increased, rather than methylglyoxal, and the effects observed may be due to increased SLG, rather than increased methylglyoxal.

Despite limitations on lab work imposed by Covid19 lockdown, we were able to perform fresh western blots on WT vs. *Alkbh7*^-/-^ samples (we have found that the MGO adduct signal degrades over time) and we have included replacement blots in Figure 3D/E, and updated the matching quantitation graphs.

Unfortunately, in April our mass-spectrometer suffered a catastrophic failure of the turbo vacuum pump, and we had difficulties in scheduling a repair due to state-wide travel restrictions. As such, it is unlikely we will be able to have functional measurements of MGO or SLG by mass-spectrometry in the near future. In lieu of a mass spectrometric assay for MGO, we attempted to measure MGO levels in WT and *Alkbh7*^-/-^ hearts using a colorimetric MGO assay kit (Biovision K500-100). However, MGO was not detectable in any of the biological samples tested. A note to this effect has been added to the Results text (subsection “Metabolomics analysis in *Alkbh7*^-/-^ confirms rewired glyoxal metabolism”).

Regarding SLG, we did previously highlight the possible role of SLG in mediating the effects of ALKBH7 loss, via lactylation of lysine residues (Discussion). We performed a western blot on WT and *Alkbh7*^-/-^ samples using a novel lactyl-lysine antibody, but simultaneously we discovered that this antibody also recognizes MGO adducts (preprint posted at https://osf.io/sba8j). In addition, another lab has raised issues on the topic of lysine lactylation (preprint at https://osf.io/kyab5). We have now modified this section of the Discussion to incorporate these novel findings.

– In line with the point addressed above, is it possible that the female mice do not display these changes in methylglyoxal metabolism and this is the reason for the sexual dimorphism? Inclusion of the female mice is a bit confusing but perhaps this is a useful "control" for the hormetic effects of methylglyoxal?

We thank the reviewers for this interesting idea. MGO adduct levels were elevated in *Alkbh7*^-/-^ males, and this drove our thinking that hormetic signaling is a mechanism underlying induction of GLO-1, which then confers protective benefits in IR injury.

Interestingly, we found that female *Alkbh7*^-/-^ mice still exhibited an elevation in GLO-1 at both the protein and activity level (new Figure 4—figure supplement 1C/D/E). However, the elevated GLO-1 activity in female *Alkbh7*^-/-^ only approached the levels seen in WT males, owing to a lower baseline GLO-1 activity in females (note – these data are split across two figures in the manuscript and supplement, so are shown collated in Author response image 1).

Underlying this lower GLO-1 activity, we also found that female *Alkbh7*^-/-^ mice did not exhibit an elevation in MGO adduct formation relative to WT (new Figure 4—figure supplement 1F/G/H). This suggests that glycative stress is blunted in females, such that female *Alkbh7*^-/-^ mice may not reach a threshold of GLO-1 activity required for protection.Although a skeptical response to these data might be “well GLO-1 is not important for protection because it is elevated in both male and female but only the males are protected”, it should be emphasized that the data in Figure 6 show that GLO-1 inhibition abrogates protection, which argues the necessity of GLO-1 for the protected phenotype. These new findings in Figure 4—figure supplement 1 are now discussed in the Results section (subsection “Loss of ALKBH7 protects the heart from ischemia-reperfusion (IR) injury”).

– Figure 6: SBB-GSH-CpE is simply a modified GSH and thus very likely to act on multiple enzymes that act on GSH and/or GSH detoxification products. While this is a commercial inhibitor "specific for GLO-1" and no characterization of the off-target effects of this compound is required, the levels of methylglyoxal, at minimum, must be measured here to demonstrate that the inhibitor is, in fact, elevating the levels of methylglyoxal.– Figure 6: are methylglyoxal-derived modifications elevated following SBB-GSH-CpE treatment?

We did measure GLO-1 activity in SBB-GSH-CpE treated hearts, and the result showing appropriate inhibition was in the text (now highlighted in the subsection “Cardioprotection in *Alkbh7*^-/-^ requires GLO-1”). As mentioned above, attempts to measure MGO levels using a colorimetric assay were unsuccessful. Regarding MGO adducts, the time of SBB-GSH-CpE treatment in the perfused heart experiments was only 10 minutes, which would be unlikely to result in elevated MGO adduct formation.

– The authors appear to have quantified α-KG in their metabolomics supplement, with no statistical changes being noted. Given the dependence of Alkbh7 on α-KG, the authors should state that this was measured, indicating no significant changes. Without demonstrating elevations in methylglyoxal and/or SLG, it is hard to conclusively argue the hormetic effects of methylglyoxal in this context.

We have added a comment addressing this point in the metabolomics Results section (subsection “Metabolomics analysis in *Alkbh7*^-/-^ confirms rewired glyoxal metabolism”). Neither α-KG nor succinate were significantly different between WT and KO, suggesting the enzyme does not contribute to the overall levels of these metabolites. This is in line with the finding that α-KGDH and SDH (complex II) activities were unaltered (Figure 2—figure supplement 2).

– The evidence for the induction of hormetic signaling in the absence of ALKBH7 is not fully supported. Specifically, the link directly tying ALKBH7 to GLO-1 is missing. The authors suggest that Nrf2 could be contributing. This would indicate that elevated levels of oxidative stress could be in play. Additionally, the authors indicate that SLG could be a link. In both cases, evidence to support either potential mechanism is not provided. Some insights are needed to support the hormetic-signaling hypothesis.

We agree, the mechanism linking loss of ALKBH7 to GLO-1 induction is still missing, and this represents a *hole* in the hypothesis that hormetic signaling is at play. We have no evidence that Nrf-2 is involved, and so have rewritten this part of the Results section to remove mention of this signaling pathway, as well as deemphasizing hormesis throughout the manuscript. The updated schematic in Figure 6D also reflects our revised interpretation of the results.

– The authors clearly show that removing glucose from the perfusate attenuates the protective effects of ALKBH7 in the setting of myocardial ischemia injury. From this data, the authors suggest that this supports a role for elevated glyoxal metabolism. However, the MGO levels should still be elevated in the hearts and more importantly GLO-1 levels should still be elevated. It seems to reason that this data would indicate a deficit in fat utilization.

While the result in question (loss of protected phenotype due to lack of glucose) could be interpreted as indicating an inability to switch to fat metabolism, and this would be consistent with previous reports on the *Alkbh7*^-/-^ mouse (Solberg et al., 2013) we now note in the text that lack of glucose resulted in no difference in baseline cardiac function, suggesting no problem in adapting to burning fat only in the knockouts (subsection “Cardioprotection in *Alkbh7*^-/-^ requires GLO-1”). This is also supported by the seahorse data showing no defect in maximal respiration on fat only. In addition, fat oxidation does not occur during ischemia, so a difference in FAO should not matter for the ischemic setting.

Nevertheless, we agree there was a lack of clarity in explaining the logic for these experiments, and it is easy to see how the paragraph describing the glucose-free perfusion experiments, with a subject heading on glyoxal metabolism and a paragraph below on GLO-1, could easily be misconstrued as supporting “a role for elevated glyoxal metabolism”.

We have re-written this section to provide more clarity on our interpretation of the glucose-only result. We have also broken out the glucose-only perfusion data and the GLO-1 inhibitor data into two separate sections, each with their own subject headings (subsections “Cardioprotection in *Alkbh7*^-/-^ requires glycolysis” and “Cardioprotection in *Alkbh7*^-/-^ requires GLO-1”) and further discussion and context for each.

– The authors provide evidence that the deficiency of ALKBH7 does not provide protection in the female heart and that with age male ALKBH7 mice lose protection against ischemic injury. It would be nice to know if the levels of MGO or GLO-1 are altered in either instance.

As described above, we have now added data on GLO-1 protein and activity and MGO adduct levels in female mice (Figure 4—figure supplement 1), and we now discuss this in the Results text (subsection “Loss of ALKBH7 protects the heart from ischemia-reperfusion (IR) injury”). We were unable to procure additional aged mice to see if similar differences were present in old vs. young, although we did provide a reference showing that GLO-1 levels decline with age (Rabbani, Xue and Thornalley, 2016).

– The authors provide compelling evidence for minimal changes in hydroxyproline levels in ALKBH7 mice at baseline. While this is indicative of basal conditions, there is not any evidence related to the activity of ALKBH7 in the setting of myocardial ischemia. If ALKBH7 is a stress-induced protein, it is possible that changes in hydroxyproline-modified proteins could become evident under these conditions.

We agree this is an intriguing possibility, but it should be noted that ALKBH7 is an α-KG dioxygenase and uses oxygen as a substrate. It is therefore unlikely that any differences in proline hydroxylation (addition of oxygen) would be seen under ischemic conditions. We have added a note about this to the Discussion.